# GMIE: a global maximum irrigation extent and central pivot irrigation system dataset derived via irrigation performance during drought stress and deep learning methods

**Fuyou Tian**[1], **Bingfang Wu**[1,2], **Hongwei Zeng**[1,2], **Miao Zhang**[1], **Weiwei Zhu**[1], **Nana Yan**[1], **Yuming Lu**[1,2], and **Yifan Li**[3,1]

[1]State Key Laboratory of Remote Sensing Science, Aerospace Information Research Institute, Chinese Academy of Sciences, Beijing 100101, China

[2]College of Resources and Environment, University of Chinese Academy of Sciences, Beijing 100049, China TS1

[3]School of Computer Science, China University of Geosciences, Wuhan 430078, China

**Correspondence:** Bingfang Wu (wubf@aircas.ac.cn)

**Abstract.** Irrigation accounts for the major form of human water consumption and plays a pivotal role in enhancing crop yields and mitigating the effects of drought. Accurate mapping of irrigation distribution is essential for effective water resource management and the assessment of food security. However, the resolution of the global irrigated cropland map is coarse, typically approximately 10 km, and it lacks regular updates. In our study, we present a robust methodology that leverages irrigation performance during drought stress as an indicator of crop productivity and water consumption to identify global irrigated cropland. Within each irrigation mapping zone (IMZ), we identified the dry months of the growing season from 2017 to 2019 or the driest months from 2010 to 2019. To delineate irrigated cropland, we utilized the collected samples to calculate normalized difference vegetation index (NDVI) thresholds for the dry months of 2017 to 2019 and the NDVI deviation from the 10-year average for the driest month. By integrating the most accurate results from these two methods, we generated the Global Maximum Irrigation Extent dataset at 100 m resolution (GMIE-100), achieving an overall accuracy of $83.6\% \pm 0.6\%$. The GMIE-100 reveals that the maximum extent of irrigated cropland encompasses $403.17 \pm 9.82$ Mha, accounting for $23.4\% \pm 0.6\%$ of the global cropland. Concentrated in fertile plains and regions adjacent to major rivers, the largest irrigated cropland areas are found in India, China, the United States, and Pakistan, which rank first to fourth, respectively. Importantly, the spatial resolution of GMIE-100 surpasses that of the dominant irrigation map, offering more detailed information essential to support estimates of agricultural water use and regional food security assessments. Furthermore, with the help of the deep learning (DL) method, the global central pivot irrigation system (CPIS) was identified using Pivot-Net, a novel convolutional neural network built on the U-net architecture. We found that there is $11.5 \pm 0.01$ Mha of CPIS, accounting for approximately $2.90\% \pm 0.03\%$ of the total irrigated cropland. In Namibia, the United States, Saudi Arabia, South Africa, Canada, and Zambia, the CPIS proportion was greater than $10\%$. To our knowledge, this is the inaugural study to undertake a global identification of specific irrigation methods, with a focus on the CPIS. The GMIE-100 dataset containing both the irrigated extent and CPIS distribution is publicly available on Harvard Dataverse at https://doi.org/10.7910/DVN/HKBAQQ (Tian et al., 2023a).

# 1 Introduction

Irrigation plays a pivotal role in mitigating the impacts of drought events (Wang et al., 2021; Wu et al., 2022). As climate change intensifies, droughts and heatwaves have become more frequent; thus, irrigation has emerged as an effective strategy to counter these extreme events and bolster the resilience of agricultural systems (McDermid et al., 2023). However, irrigation represents a significant human intervention in the global water cycle, as it accounts for 67 % of global freshwater withdrawal and 87 % of total water consumption (Wu et al., 2022). Therefore, accurate information about irrigation is critical for both crop monitoring and water resource management purposes (Wu et al., 2023b; Tian et al., 2022). However, the highest available resolution for existing irrigation maps remains within a range of 500 m to 10 km (Nagaraj et al., 2021; Siebert et al., 2005, 2013). This resolution is insufficient to support effective crop condition monitoring and sustainable water resource management at the subbasin level (Zhang et al., 2022b; Xie and Lark, 2021).

Traditionally, two methods have been used for generating gridded irrigation maps. The first method involves the allocation of statistical data that uses specific indicators such as land cover area, peak normalized difference vegetation index (NDVI) values, and irrigation potential indices (Zhu et al., 2014; Pervez and Brown, 2010; Zajac et al., 2022). For example, the Food and Agriculture Organization (FAO) applied this approach to produce the Global Map of Irrigation Area (FAO-GMIA) from 1995 to 2005 at a 10 km resolution; this renowned irrigation map is widely applied in global water resource management (Siebert et al., 2015). At the national scale, several irrigation maps for China have been produced with resolutions ranging from 500 to 1000 m; these maps primarily utilize data from the Chinese Statistical Yearbook (Zhu et al., 2014; Zhang et al., 2022c). For the United States, Pervez and Brown (2010) developed the MODIS Irrigated Agriculture Dataset for the United States (MIrAD-US) with a resolution of 250 m. Zajac et al. (2022) produced the European Irrigation Map for 2010 (EIM2010), albeit with a coarser 10 km × 10 km resolution. Importantly, the accuracy of irrigated cropland maps generated through these methods relies heavily on the representativeness of the spatial allocation indicators and the precision of the statistical data. The indicators used to allocate irrigation areas to each grid often fail to capture the precise distribution of irrigated cropland, especially in humid regions (Pervez and Brown, 2010). As a result, achieving higher-resolution irrigation maps using this approach can be challenging. Furthermore, due to variations in terrain types and irrigation techniques, census data may underestimate the actual irrigation area (Zhang et al., 2022b). Furthermore, data from different departments may exhibit discrepancies owing to differing statistical criteria. For example, in 2010, the reported irrigation area in California differed by more than 10 % between the US Geological Survey and the state's Department of Water Resources (Meier et al., 2018).

Scholars have sought to independently derive irrigated cropland using spectral signatures (Thenkabail et al., 2009; Salmon et al., 2015). The peak values in time-series vegetation indices can serve as indicators of crop water stress, biomass, and chlorophyll content. Given that irrigated crops typically exhibit reduced water stress and elevated chlorophyll content, disparities in peak vegetation index values can be harnessed to differentiate between irrigated and rainfed croplands. Commonly employed vegetation indices for this approach include the NDVI, greenness index (GI), land surface water index (LSWI), chlorophyll vegetation index (GCVI), and enhanced vegetation index (EVI) (Shahriar Pervez et al., 2014; Lu et al., 2021; Chen et al., 2018; Xiang et al., 2019; dela Torre et al., 2021). The discrimination between irrigated and rainfed croplands is typically accomplished through thresholding or decision tree classification and relies on selected vegetation indices. Nevertheless, importantly, vegetation indices may not entirely capture crop water stress, leading to subtle differences in peak vegetation indices and complicating the mapping of large-scale irrigated farmland.

To improve the delineation of irrigated cropland, supervised classification models incorporate climate variables and environmental factors, such as precipitation, temperature, surface temperature, and terrain (Salmon et al., 2015). For instance, Thenkabail et al. (2009) combined AVHRR vegetation index time series, precipitation data, elevation information, and vegetation cover maps as inputs to a decision tree classifier, resulting in the creation of the first global irrigated area map (IWMI-GIAM) at a 10 km resolution based on remote sensing data. Salmon et al. (2015) employed MODIS vegetation indices and 19 climate variables to produce the Global Rainfed and Irrigated Cropland map (GRIPC-500) for 2005 at a resolution of 500 m.

In recent years, the mapping of irrigated croplands at national and regional scales has undergone significant advancements due to the availability of extensive meteorological and remote sensing data stored in Google Earth Engine (GEE) (Zhang et al., 2022b; Deines et al., 2019; Xie et al., 2019; Xie and Lark, 2021). Xie et al. (2021) developed a random forest model that incorporates a wide array of variables, including environmental factors (precipitation, Palmer drought severity index, soil moisture, aridity index, land surface, and air temperature), vegetation indices (NDVI, NDWI, GCVI, WGI, and AGI), and ground irrigation samples. This model achieved an impressive 30 m resolution irrigation dataset for the United States (LANID). Subsequently, Zhang et al. (2022a) applied this methodology to generate an irrigated cropland map for China from 2000 to 2019 with a resolution of 500 m (IrriMap_CN). In the same year, Zhang et al. (2022c) enhanced the resolution of the irrigation cropland distribution map for China to 250 m. However, this method heavily relies on sample data,

and the spatial representativeness of these irrigation and rain-fed samples directly influences the accuracy of the results (Zhang et al., 2022b). Collecting ground sample points is a labour-intensive and time-consuming process, and ensuring their spatial representativeness across larger areas, including at a global scale, poses considerable challenges (Zhang et al., 2022c, d; Tian et al., 2022).

Though various irrigation maps exist at global and national scales, many of these maps suffer from either very low spatial resolution or outdated information, as outlined in Table 1 (Dari et al., 2023). Among these data, the Landsat-derived Global Rainfed and Irrigated-area Product (LGRIP30) is a high-resolution irrigated cropland with an overall accuracy of 86.5 % using advanced machine learning algorithms, which was released on February 2023 and is available through NASA's Land Processes Distributed Active Archive Center (LP DAAC) (Teluguntla et al., 2023). The LGRIP30 data indicate a total global net irrigated area (TGNIA) of 0.71 Gha among all cropland area of 1.80 Gha of croplands, meaning the irrigation proportion was approximately 39.44 %, suggesting a relative high proportion compared to exiting result (Thenkabail et al., 2009; Siebert et al., 2015). While some high-resolution irrigation maps are annually updated, they are typically applicable only at a national level (Zhang et al., 2022b; Xie et al., 2021). Thus, the challenge of generating a higher-resolution and up-to-date global irrigated cropland map via supervised methods persists.

An additional significant issue is the phenomenon of "mixed pixels" in MODIS data, which is particularly pronounced in regions with fragmented croplands, such as farmlands in southern China and Africa, where agricultural fields are often smaller than one MODIS pixel (0.25 ha TS2) (Zhang et al., 2022a). Consequently, global irrigation maps with higher resolution are urgently needed to support both water resource management and food security assessments.

Inspired by the fundamental purpose of irrigation, which is to alleviate the impact of drought, we introduced the Global Maximum Irrigated Extent with 100 m resolution (GMIE-100) dataset. This dataset leverages irrigation performance during periods of drought stress. When drought conditions prevail, disparities in crop conditions, as indicated by the peak NDVI values, become more pronounced between irrigated and rainfed farmlands. This amplification enables the precise identification of irrigated farmland across most regions while also reducing the number of required training samples (Wu et al., 2023a).

Furthermore, considerable variations in irrigation efficiency are apparent among different irrigation types, with central pivot irrigation systems (CPISs), which achieve an efficiency rate exceeding 80 % and are the predominant global sprinkler irrigation method (Tian et al., 2023b). In contrast, gravity-flowing irrigation methods, while widespread, exhibit a comparatively lower efficiency rate of approximately 60 % (Waller and Yitayew, 2016). Despite the important role of irrigation in agriculture, few studies have been dedicated to the remote sensing identification of various irrigation types, indicating a notable gap in scientific exploration. Notably, the unique circular configuration of CPISs facilitates their visual interpretation from satellite imagery, presenting an avenue for enhanced monitoring and analysis through remote sensing technologies. The advent of deep learning (DL) has opened avenues for the classification of types of irrigation methods based on distinctive spatial patterns, such as the CPIS. In this study, Pivot-Net, a shape-attention neural network designed for CPIS identification in satellite imagery, was used, and a global CPIS dataset (GCPIS) was generated to estimate the proportion of types of irrigation methods for the CPIS.

## 2 Materials and methods

Taking inspiration from the fundamental purpose of irrigation, our aim is to identify periods of drought stress to highlight disparities in crop conditions between irrigated and rainfed croplands. We began by utilizing the 65 monitoring and reporting units (MRUs) established by CropWatch (Wu et al., 2015; Gommes et al., 2016). These MRUs, which account for factors such as crop types, agricultural potential, and environmental conditions, served as the foundation for dividing global cropland into 110 irrigation mapping zones (IMZs). The first-level 65 agroecological zones provide a broad global overview. To address limitations in representing water stress and irrigation within zones, we introduced a more detailed classification, creating second-level agroecological zones based on arid indices, water availability, soil types, and landforms. Ultimately, we utilized 110 IMZs as the foundational units for determining the specific timing of drought stress, as illustrated in Fig. 1. This comprehensive approach enabled us to capture and amplify the distinctions in crop conditions between irrigated and rainfed croplands. Irrigated cropland is defined as agricultural land that benefits from human interventions and is equipped with irrigation infrastructure, including facilities like canals and central pivot systems (Salmon et al., 2015; Meier et al., 2018). This definition includes areas that receive irrigation at any time during the season, regardless of whether they are irrigated in every season or not.

The general framework for detecting drought stress and evaluating crop conditions in irrigated and rainfed cropland is illustrated in Fig. 2. The study was inspired by the purpose of irrigation, i.e. that it mitigates the effect of water stress. Basically, we assume that water stress can be regular or irregular. If there are crops during the dry season, the irrigation should occur regularly. Otherwise, irrigation is just complementary to rainfall in extremely dry years, which means irrigation is irregular. For regular irrigation, we could detect vegetation signal in the dry season (DM-NDVI) when precipitation cannot meet water demand for crops. For irregular irrigation, we compare the NDVI in extremely dry years to the 10-year av-

**Table 1.** List of existing irrigation maps at the global or national scale.

| Dataset | Coverage | Spatial resolution | Time | Method summary | Reference |
|---|---|---|---|---|---|
| Global Irrigated Area Map (IWMI-GIAM) | Global | 10 km | 2000 | Uses decision tree classifier with vegetation index & environmental data as input | Thenkabail et al. (2009) |
| Global Map of Irrigation Area (FAO-GMIA) | Global | 10 km | 1995/2000/2005 | Allocates census data based on land cover area | Siebert et al. (2015) |
| Global Rainfed, Irrigated and Paddy Croplands (GRIPC-500) | Global | 500 m | Single map 2005 | Includes climate variables and environmental factors in a decision tree classifier | Salmon et al. (2015) |
| Global Food-Support Analysis Data (GFSAD) | Global | 1 km | 2010 | Created using multiple input data including satellite, climatic, and census data | Thenkabail et al. (2012) |
| Landsat-derived Global Rainfed and Irrigated-Cropland Product at nominal 30 m of the World (USGS-LGRIP30) | Global | 30 m | 2015 | Landsat-derived global rainfed and irrigated cropland product within cropland extent | Teluguntla et al. (2023) |
| Landsat-based Irrigation Dataset (LANID) | United States | 30 m | 1997–2017 | Random forest model based on environmental variables & vegetation indices | Xie et al. (2021, 2019); Xie and Lark (2021) |
| Annual irrigation maps across China (IrriMap_CN) | China | 500 m | 2000–2019 | Random forest with remote sensing index and environmental index | Zhang et al. (2022b) |
| Remotely sensed high-resolution irrigated area in India | India | 250 m | 2000–2015 | NDVI series in decision tree method | Ambika et al. (2016) |

erage level and calculate the deviation (NDVI$_{dev}$) to determine whether it is irrigated or not. To determine whether a region has regular or irregular irrigation, we used both of these indicators and chose the method with the higher accuracy.

Then, with the support of the DL model, a CPIS identification model focused on circular shapes was trained and applied to the entire world to generate global CPIS distribution data. The extent of the CPIS was recognized as the extent of irrigation used to update the global extent of irrigation. Finally, we estimated the proportion of irrigation types of CPIS within irrigated cropland.

## 2.1   Input data

In this research, the distribution of rainfall on a global scale plays a pivotal role in determining the necessity for crop irrigation. The focus of this study was the 10-year period from 2010 to 2019, and the aim was to identify the driest year within this time frame. Two distinct sources of precipitation data were utilized:

(a)  Tropical Rainfall Measuring Mission (TRMM) data from the TRMM collection TRMM/3B43V7, which provides monthly precipitation estimates, were employed for geographical areas ranging from 50° S to 50° N. This data source offers insights into precipitation patterns within this specific region.

(b)  Global Land Data Assimilation System (GLDAS) data for precipitation were used for areas outside the 50° S to 50° N range, as GLDAS provides information on precipitation in regions beyond the tropical band.

Additionally, the evapotranspiration product, MOD16A2.006, which was introduced by Mu et al. in 2013 (Mu et al., 2013) TS3, was utilized. This product can determine the water surplus during the driest months within each IMZ. The MOD16A2.006 dataset is characterized by an 8 d composite time frame and a pixel resolution of 500 m. It is derived from the Penman–Monteith equation and incorporates both daily meteorological reanalysis data and remotely sensed data products from MODIS. This comprehensive dataset aids in the assessment of water availability and evapotranspiration dynamics during critical dry periods.

The 30 m spatial resolution NDVI data from the Landsat sensors Thematic Mapper (TM), Enhanced Thematic Mapper Plus (ETM+), and Thermal Infrared Sensor (OLI-TIRS) on board Landsat-5, Landsat-7, and Landsat-8, respectively,

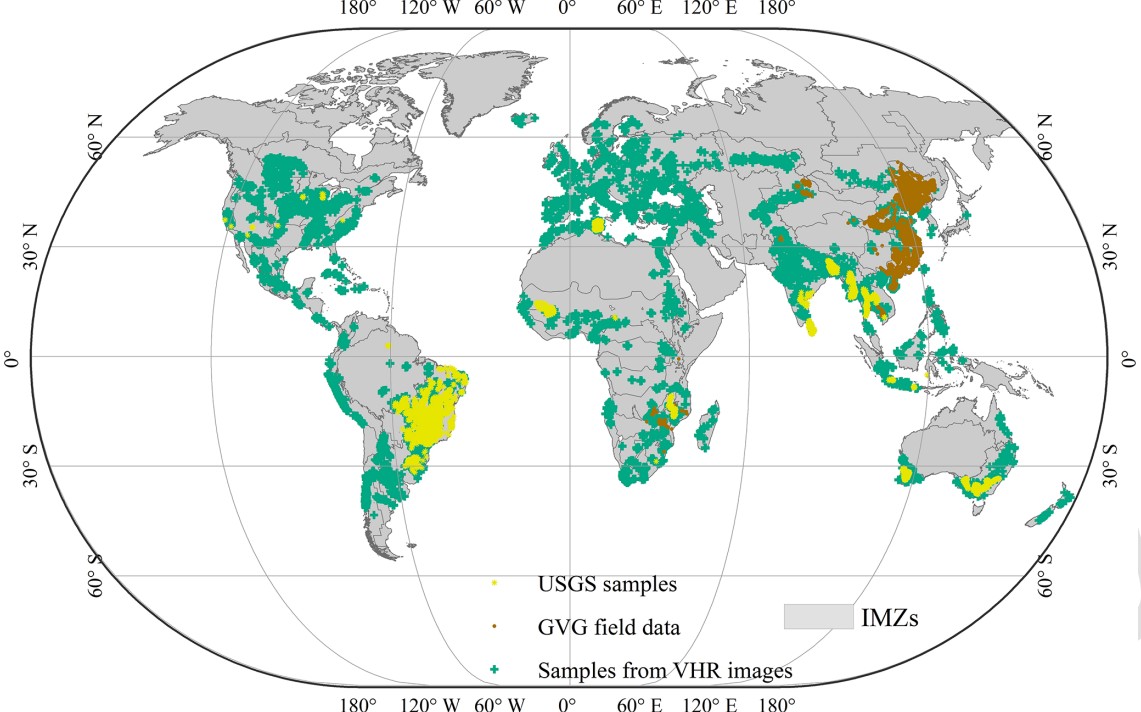

**Figure 1.** Samples of irrigated, rainfed, and central pivot irrigation system (CPIS) from multiple sources and mapping units for irrigation mapping and CPIS identification. GVG is the GPS, Video, GIS system for collecting field data. VHR means very high resolution. IMZs means irrigation mapping zones.

were used in Google Earth Engine (GEE) (Gorelick et al., 2017) to differentiate irrigated and nonirrigated areas across various IMZs during a specific period. The NDVI data were masked using the cloud and water mask in the flag file and rescaled into the same range between −1 and 1.

## 2.2 Sample data

Acquiring irrigation samples on a global scale presents an enormous challenge due to significant labour and cost requirements, primarily attributable to the extensive geographic scope. To globally classify irrigated and nonirrigated cropland, a single dataset of adequately representative samples is needed; however, such a dataset is currently unavailable. The scarcity of irrigation datasets tailored to specific crop types hinders precise differentiations between irrigated and nonirrigated croplands. In most countries, except for India, China, and Pakistan, the area allocated to irrigated croplands constitutes a relatively minor fraction of the total cultivated area. This paucity of representation poses challenges in amassing a substantial sample size suitable for classification purposes. Contemporary irrigation maps often have coarse spatial resolutions, which curtail their efficacy in generating precise samples for classification endeavours. To overcome these limitations and establish a robust sample dataset, an integrative methodology was employed. This approach entailed the fusion of data originating from three independent sources, facilitating a more comprehensive and accurate appraisal of global irrigated and nonirrigated croplands.

The first source comprises field data points collected using the GVG (GPS, Video, GIS) application in China (surveyed from 2010 to 2019), Cambodia (in 2019), Ethiopia (from 2018 to 2019), Zambia (from 2016 to 2019), Mozambique (from 2016 to 2019), and Zimbabwe (from 2016 to 2019). This application serves as a comprehensive field data collection system that integrates GPS for precise positioning, a video for capturing geo-tagged photographs, and a GIS system for managing geographic information (Wu et al., 2023a, 2020), which can be downloaded via https://gvgserver.cropwatch.com.cn/download (last access: 17 February 2025). By conducting observations of irrigation infrastructure, including irrigation canals, reservoirs, lakes, rivers, and irrigation wells, and through interactions with farmers, we were able to determine the types of irrigation in the fields. Additionally, irrigation was applied for certain crop types, such as winter wheat in the North China Plain, cotton in Xinjiang, and vegetable and tomatoes in most provinces. Meanwhile, irrigated crops usually appear greener and lusher compared with nearby crops. If it cannot be distinguished following the above characteristics, the inquiry of local farmers could give the answer. The collected dataset comprises a total of 78 338 sample points, including 36 809 rain-

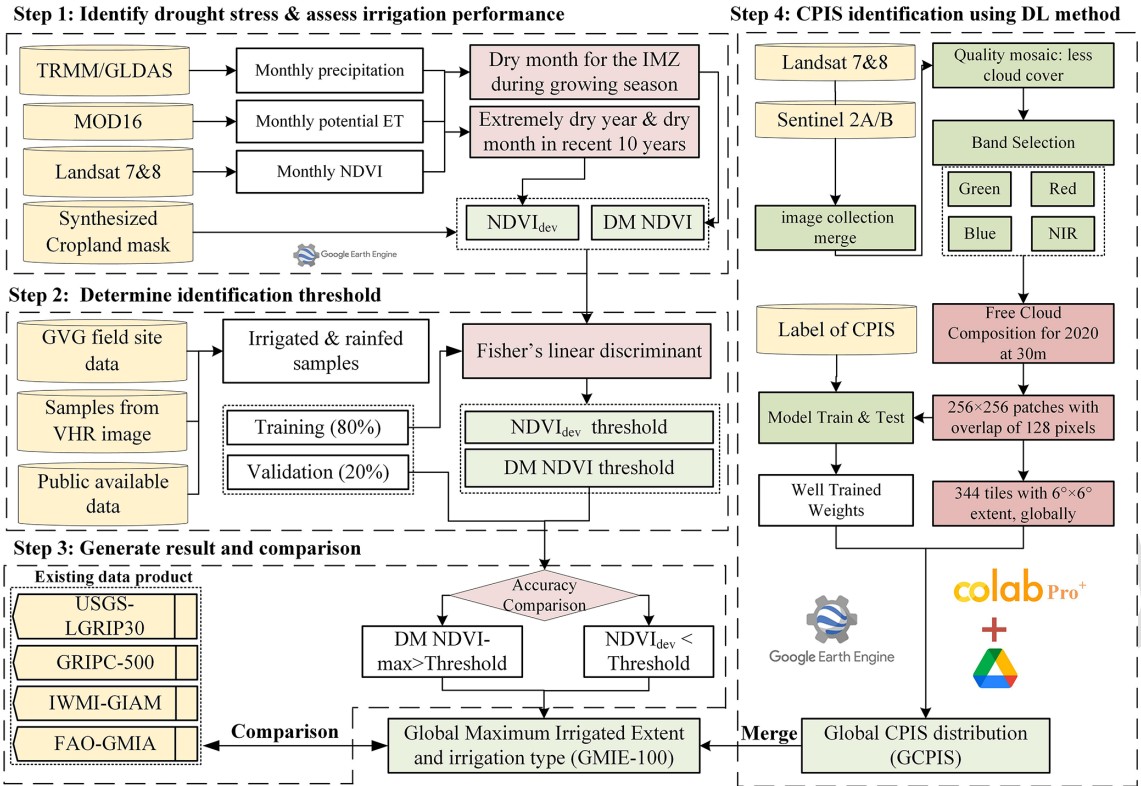

**Figure 2.** Flow chart of GMIE-100 with a typical irrigation type of CPIS. GVG is the GPS, Video, GIS system for collecting field data. VHR means very high resolution. IMZs means irrigation mapping zones. $NDVI_{dev}$: NDVI deviation in extremely dry years with the 10-year average level. DM-NDVI: NDVI in the dry season.

fed samples and 41 529 irrigation samples, with the majority of these points located in China, totalling 72 224 points.

The second data source consists of validation points collected as part of the Global Food Security Analysis Data 30 (GFSAD30) project, which is made available to the public through the website https://croplands.org/app/data/search (last access: 17 February 2025). This project is a collaborative effort involving the United States Geological Survey (USGS), various universities, research institutions, and companies such as Google. These sample points were collected or derived as part of the project's objective to support global food security analysis at a 30 m spatial resolution. Some samples were collected via field surveys conducted using mobile applications. Others were derived from interpretations of remote sensing imagery, such as MODIS and Landsat TM data, crop-specific thematic maps, foundational geographic data (e.g. road networks), and other geospatial information (e.g. elevation data layers). The dataset encompasses a total of 17 076 sample points, comprising 3000 rainfed points and 14 076 irrigated points. The majority of these points are located in Brazil (13 368), Australia (2192), Thailand (393), and Tunisia (389).

The third supplementary data source involved the acquisition of samples through visual interpretation of very high

resolution (VHR) images available in GEE. The following irrigation points were selected based on identifiable irrigation infrastructure: (1) central pivot irrigation systems, which are easy to identify due to their shapes; (2) clearly visible irrigation systems, which are clearly visible on VHR images; (3) rain-deficient cultivated areas, which are areas classified as cropland with insufficient rainfall but exhibiting NDVI values indicating vegetation presence and annual growth rings; and (4) high vegetation signals during dry seasons, identified by elevated vegetation indices during these periods. The United Nations Food and Agriculture Organization's Global Map of Irrigation Areas (FAO GMIA) (Siebert et al., 2013) and the World Heritage Irrigation Structures (WHIS) list (https://www.icid.org/icid_his1.html, last access: 17 February 2025) were used as reference sources. The FAO GMIA's Irrigation Areas of Interest (AEI) and WHIS listings were consulted to identify irrigation areas. Rainfed irrigation points were selected based on FAO GMIA's criteria. If a region lacked any irrigation infrastructure and the AEI value from the FAO GMIA was zero, the area was classified as a rainfed irrigation sample.

Figure 1 illustrates a total of 115 379 sample points. In total, 80 % of these data, or 92 303 points (comprising 37 650 rainfed and 54 653 irrigated points), were employed for train-

ing or calibrating the threshold. The remaining 20 %, or 23 076 points (comprising 10 892 rainfed cropland points and 12 184 irrigated points), were used for result validation.

## 2.3 Land cover and cropland datasets

In this research, we delineated irrigated croplands within the extent of cropland. The definition of cropland was the same as that of the Joint Experiment of Crop Assessment and Monitoring (JECAM) network for the Group on Earth Observations Global Agricultural Monitoring Initiative (GEOGALM), which defines the land used for seasonal crops (sowed/planted and harvested at least once within 12 months), such as cereals, root, and tuber crops; for oil crops; and for economically significant crops, such as sugar, vegetables, and cotton (Waldner et al., 2016). Additionally, the land occupied by greenhouses was considered cropland. To achieve comprehensive global cropland coverage, the synthesized data were obtained from 16 recent national and regional datasets spanning 2015–2019, which were supplemented by two global satellite-derived land cover datasets, as listed in Table 2. In this study, all land cover classes that met the cropland definition were consolidated into a single category labelled "cropland". On the other hand, various non-vegetation land cover classes (e.g. urban or water) and vegetated classes (e.g. forest or grasslands), including agricultural categories (e.g. permanent crops, cultivated rangeland, and grassland), were amalgamated into one class as "noncropland". The cropland mask at a 30 m resolution could be obtained from the International Research Center of Big Data for Sustainable Development Goals via https://data.casearth. cn/thematic/cbas_2022/158 (last access: 17 February 2025). These data integrated more than 10 cropland datasets including global cropland products, FROM-GLC (Yu et al., 2013) and GFSAD30 (Thenkabail et al., 2021), and national and regional datasets, such as ChinaCover (Wu et al., 2017, 2024), Cropland Data Layers (Boryan et al., 2011), Agriculture and Agri-Food Canada Annual Crop Inventory (Fisette et al., 2013; McNairn et al., 2009), and MapBiomass (do Canto et al., 2020). More information about this cropland mask can be found in the Supplement. These data have been utilized for their extensive validation by local experts, leading to their high precision in mapping cropland (Wu et al., 2023a). The overall accuracy of this cropland was 89.4 %. Moreover, this mask has also been employed in other studies to map global crop intensity (Zhang et al., 2021a).

## 2.4 Irrigation mapping method

### 2.4.1 Identifying the dry months and dry years

The cumulative yearly rainfall and monthly rainfall ($P$) for 2010–2019 were calculated from the TRMM dataset for all the IMZs via GEE. Simultaneously, monthly potential evapotranspiration (PET) data for the same time were derived from the MOD16A2.006 product in GEE. The monthly water surplus ($P - PET$) was calculated by subtracting the monthly P and the monthly PET.

Within the growing seasons of 2017–2019, we identified the dry months by pinpointing the lowest differences between the monthly $P$ and PET. Additionally, we determined the driest year from 2010–2019 based on the lowest annual $P$, and the corresponding driest month was identified as the month with the lowest $P - PET$ value during the driest year within the growing season.

### 2.4.2 Identifying thresholds of NDVI and NDVI deviation

Irrigated cropland is characterized as cropland subjected to human interventions and equipped with irrigation infrastructure, including systems such as canals and CPISs (Wu et al., 2023a). The specific threshold for distinguishing between irrigated and nonirrigated cropland varies across IMZs. The threshold for each IMZ was determined by training samples through visual interpretation of very high resolution images from Google Earth.

For each IMZ, the maximum NDVI was calculated within the cropland extent during dry months (NDVImax-DM) using Landsat-8 images in Google Earth Engine to detect vegetation signals. In regions where regular irrigation is necessary, irrigated cropland can be mapped annually. However, to avoid missing fallow land based on the results of a single year, irrigated croplands were identified through the NDVI threshold over a 3-year period from 2017 to 2019.

For regions with ample rainfall, drought stress may not be a concern. Hence, satellite data spanning the 2010–2019 period were utilized to identify the crop conditions during extreme drought events. The NDVI deviation (NDVI$_{dev}$) was calculated for the driest month of the driest year from 2010–2019 for the cropland pixels according to the following formula:

$$\text{NDVI}_{dev} = \frac{\text{NDVI}_{max-DriestM} - 10\text{YNDVI}_{DM}}{10\text{YNDVI}_{DM}}, \quad (1)$$

where NDVI$_{max\text{-}DriestM}$ is the maximum NDVI value in the driest month over 10 years, and 10YNDVI$_{DM}$ is the monthly average NDVI in the same month.

For each IMZ, the midpoint value for a cropland pixel was determined from the irrigated and nonirrigated training points via Fisher's linear discriminant (Duda et al., 2012):

$$\text{Nmidpoint} = \frac{N_{irrigated} + N_{nonirrigated}}{2}, \quad (2)$$

where $N_{irrgated}$ and $N_{nonirrigated}$ represent the mean values of the NDVI or NDVI$_{dev}$ at irrigated and nonirrigated points, respectively.

For each IMZ, the Nmidpoint of the NDVI value and NDVI$_{dev}$, serving as the threshold value, was calculated using irrigated and rainfed samples. Subsequently, pixels exhibiting an NDVI exceeding their specific threshold values

for dry months or an NDVI$_{dev}$ below the threshold during the driest month of the driest year were designated irrigated; otherwise, the pixels below the threshold were classified as nonirrigated.

The final threshold value was determined by selecting the NDVI or NDVI$_{dev}$ threshold that yielded the highest overall accuracy in distinguishing irrigated cropland in the validation samples. Subsequently, the chosen threshold value for either the NDVI or NDVI$_{dev}$ of the IMZ was applied to the respective pixels, which were accepted as the final results. If the maximum NDVI value in dry months achieved higher accuracy for identifying irrigated cropland, the corresponding region usually needs regular irrigation and thus is labelled as region irrigation regular (RIR). Otherwise, the region needs irrigation only occasionally for some years and thus is labelled as region irrigation occasional (RIO).

Taking IMZ C48, primarily situated in Pakistan, as an example, Fig. 3a illustrates the monthly NDVI profile for the year 2017 within Pakistan (IMZ C48, south Asia, Punjab to Gujarat). It is evident that the discrepancy in NDVI values between irrigated and nonirrigated crops remained marginal for the majority of the months in 2017. However, in February 2017, during a period of drought stress characterized by a meager precipitation of 4.4 mm or a precipitation-to-evapotranspiration ratio of 0.02, the disparity in NDVI values became notably more pronounced and distinguishable. Consequently, the optimal NDVI threshold of 0.44 was ascertained to be the most suitable for discriminating irrigated from nonirrigated regions, as depicted in Fig. 4b.

For the RIO, IMZ C58 was chosen as an example. Figure 3d and f illustrate the monthly NDVI profiles for the extreme drought year of 2012, the 10-year average NDVI value, and the NDVI deviation of the extreme drought year from the 10-year average. The comparison revealed that rainfed cropland exhibited more substantial fluctuations in the NDVI than did irrigated cropland. Consequently, the NDVI$_{dev}$ (NDVI deviation) during severe drought or extremely arid conditions was employed to differentiate irrigated cropland from other categories. The NDVI$_{dev}$ midpoint was established as 0.12 following Eq. (2).

By combining these two categories of irrigated cropland, we created a comprehensive global irrigation map. For further detailed information, please refer to Wu et al. (2023a). Originally, the Global Maximum Irrigated Extent (GMIE) dataset was established at a 30 m resolution, featuring a binary classification into irrigated and rainfed cropland. This resolution was determined by the availability of cropland masks and NDVI data, both of which are at the 30 m scale. However, the extent of irrigation may vary due to crop rotation and fallow cropland, which are clearly observable at a 30 m resolution and impact the extent of irrigated cropland. We calculated the irrigated cropland proportion within 100 m × 100 m to reduce these effects. The GMIE-100 dataset ranges from 0 to 1, with a no-data value set at −99.

## 2.5  CPIS identification

Inspired by the spatial attention gate, four attention blocks were incorporated into the connections between downsampling and upsampling within the U-Net architecture (Fig. 4). Pivot-Net incorporates four spatial attention gates to effectively capture information about the round shape of the CPIS. To enhance the model's ability to understand shape-related intermediate features during boundary detection and segmentation tasks, a multitask learning approach was employed to train the model. This approach integrates pixel-wise segmentation and boundary prediction as integral components of Pivot-Net's learning objectives. This method was successfully applied to identify CPISs for the whole of the United States (Tian et al., 2023b).

We generated composite, cloud-free satellite data by utilizing optical images from Sentinel-2 and Landsat-8 for each tile within GEE from March to August 2020. All exported data from GEE were stored in Google Drive. The world was divided into 345 tiles of 6° × 6°, 23 of which were annotated manually (Fig. 5). In total, 80 % of all the CPIS labels or 9140 patches with 256 × 256 pixels were used for training the model, and the remaining 20 % of the CPIS labels or 2284 patches were used for accuracy validation.

Subsequently, we transferred the trained model, which was stored on a local high-performance computer, to Google Drive. By employing the robust computational capabilities of Google Colab Pro+ (https://colab.research.google.com/, last access: 17 February 2025), which seamlessly accesses satellite data in Google Drive, we applied the well-trained Pivot-Net model across all tiles. The satellite data were partitioned into 256 × 256 patches with a 128-pixel overlap (stride = 128 pixels). The final prediction was determined by selecting the maximum prediction probability within the overlap region.

## 3  Results and discussion

### 3.1  Spatial pattern of irrigated cropland and GCPIS

The spatial distribution of GMIE-100 is shown in Fig. 6. The GMIE-100 revealed that the maximum extent of irrigated cropland is $403.17 \pm 9.82$ Mha (million hectares), which accounts for $23.4 \% \pm 0.6 \%$ of the global cropland, equivalent to 1724.08 Mha. This figure surpasses the total area equipped for irrigation reported by FAOSTAT for 2000–2008 (307.60 Mha) (Siebert et al., 2013) and closely aligns with the irrigated area estimated by IWMI–GIAM (406.40 Mha, representing 19.5 % of global cropland in 2000) (Thenkabail et al., 2009). India (94.85 Mha, representing 50.4 % of cropland) has the largest area of irrigated cropland in the world, with China (85.16 Mha, 50.0 % of cropland) and Pakistan (18.04 Mha, 80.2 % of cropland) ranking second and fourth, respectively. In addition, the United States (26.54 Mha, 15.5 % of cropland) ranks third globally in terms

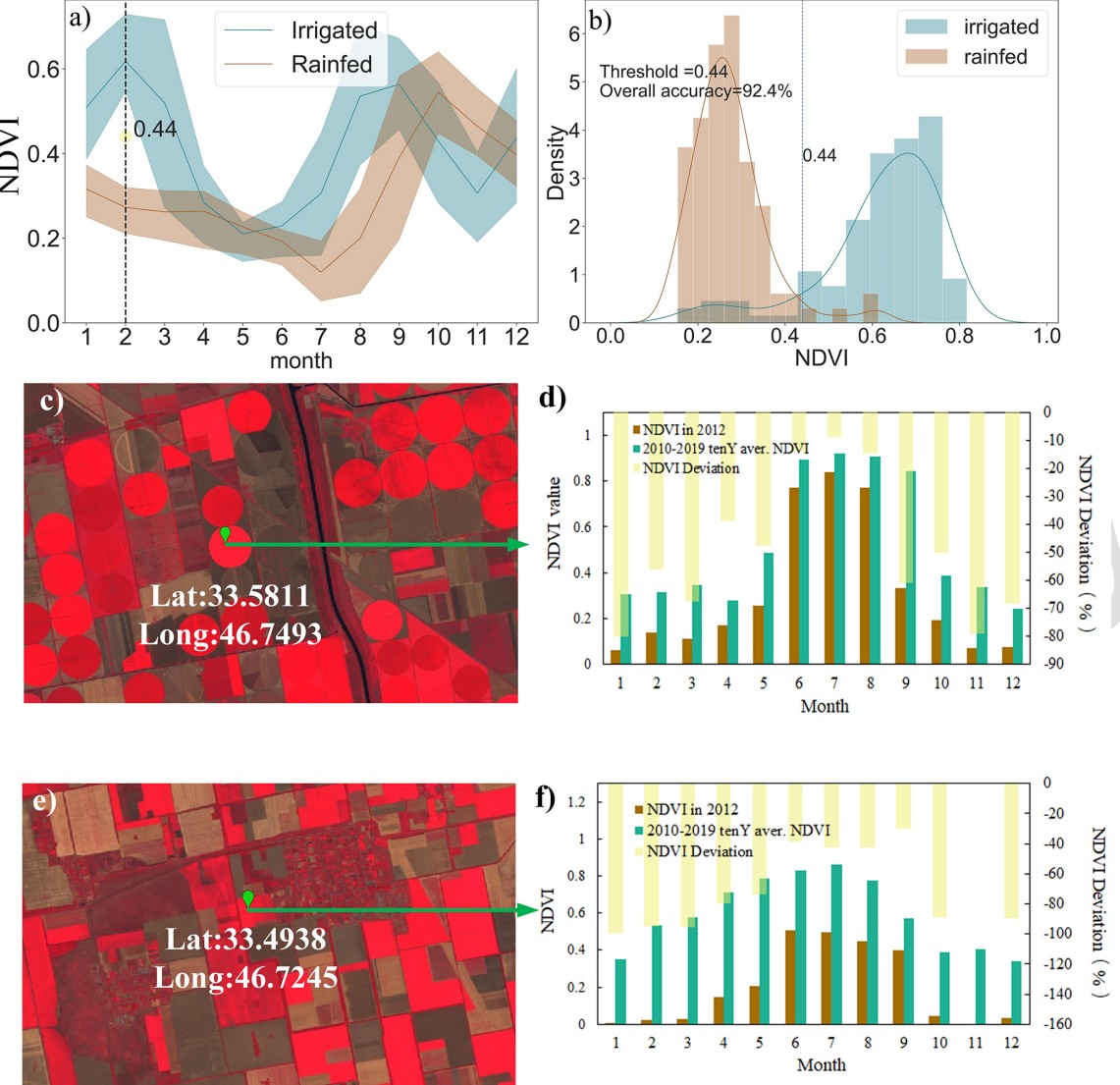

**Figure 3.** NDVI profile in 2017 **(a)**; NDVI histogram in February 2017 **(b)** (Pakistan IMZ C48 as an example); monthly NDVI in an extremely dry year (2012), 10-year average NDVI, and $NDVI_{dev}$ for typical central pivot irrigated cropland **(c, d)** and rainfed cropland **(e, f)** in southern Ukraine (IMZ C58). The background images in **(c)** and **(e)** are Landsat-8 images. Panels **(c)** and **(e)** are credited to the U.S. Geological Survey.

of irrigated cropland. For the remaining countries, less than 10 Mha of cropland is irrigated.

The irrigated cropland is notably concentrated in regions characterized by expansive plains and proximity to rivers. These flat and river-proximal areas are well suited for irrigation due to easy access to water resources (Jianxi et al., 2015; Wu et al., 2021). In fact, a substantial portion of the global irrigated cropland, encompassing 224 Mha, or 55.6 % of the total irrigated cropland, is situated in such plain regions. Prominent examples include the Ganges Plain, the Indus Plain, and the North China Plain, all of which host significant expanses of irrigated cropland. Nevertheless, despite their close proximity to water sources, there are areas where the proportion of irrigated land remains low. For instance, regions such as the Danube estuary in Romania exhibit an irrigation proportion of 3.65 %, despite experiencing high annual food production variability (Wriedt et al., 2009). Similarly, the Zambezi basin, which encompasses countries such as Zambia (4.1 %) and Mozambique (4.2 %), struggles with food insecurity despite its access to water resources.

Apart from plains, oases within arid zones represent a significant category of regions with extensive irrigated cropland. These areas are distinctive due to their limited precipitation but abundant sunlight and heat resources (Chen et al., 2023b). In oases, the availability of irrigation is crucial for crop survival. Approximately 31 Mha of irrigated cropland

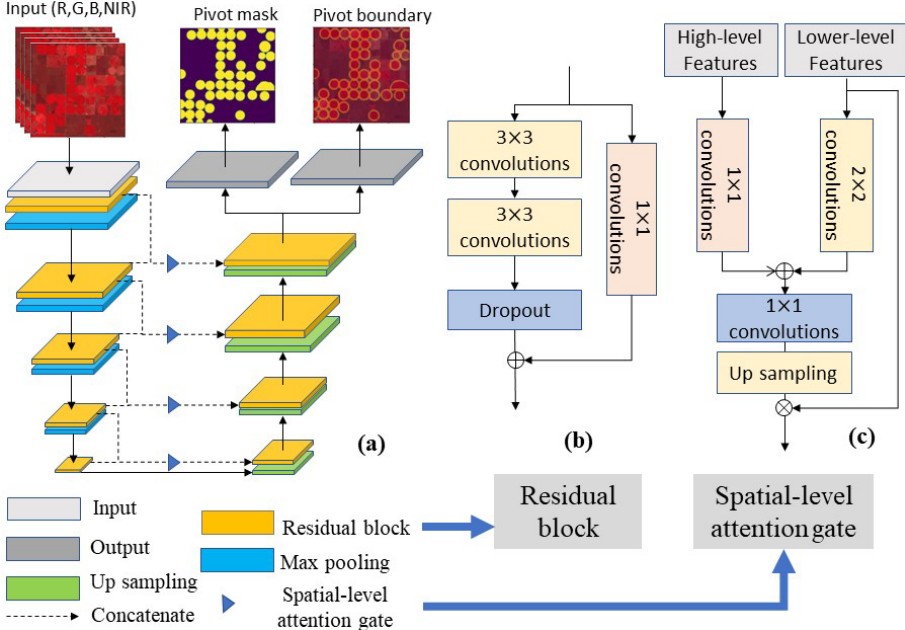

**Figure 4.** Architecture of the shape-attention Pivot-Net (Tian et al., 2023b).

is situated within arid zone oases, constituting 7.7 % of the total irrigated cropland. Well-known oasis agricultural regions across the world include the Nile basin and the delta region in Egypt, the California Valley in the United States, and Xinjiang in China. These areas thrive due to their irrigation practices, which enable the productive use of scarce water resources amid arid conditions (Cui et al., 2024).

The distribution of irrigated cropland exhibits distinct patterns when examined from both latitude and longitude perspectives. Along the latitude, we observe exceptionally high irrigation proportions around the 30° N latitude, which encompasses regions along the lower Yangtze River, Ganges River, Indus River, and Nile River. These river basins are characterized by dense concentrations of irrigated cropland, owing to the availability of water resources from these major river systems (Nagaraj et al., 2021). On the other hand, when assessing irrigation proportions along the longitude, we observe elevated levels of irrigation between 60 and 120° E. This longitudinal span encompasses prominent regions such as the Indus-Ganges Plain and the North China Plain, which are renowned for their high levels of irrigated agriculture.

For the CPIS worldwide, the spatial pattern is depicted in Fig. 7. The total area of the CPIS is estimated to be $115\,192.2 \pm 100.0\,\text{km}^2$, comprising 2.9 % of the total irrigated area. The area in Chen's research is $107\,232.8\,\text{km}^2$ (Chen et al., 2023a) in global arid regions. The CPIS is mainly distributed in the high plain aquifers (HPAs), including north Texas, Kansas, and Nebraska; southern Brazil; South Africa; and the Middle East region. Along the longitude, the CPIS proportion is high from 90 to 120° W, which matches the range of HPAs, while the CPIS proportion is relatively apparent between 30 and 60° N with latitude.

The distributions of irrigated cropland and CPIS proportions across the six continents are depicted in Fig. 8a. Asia has the most irrigated area, covering 273.79 Mha, with an irrigation proportion of 39.3 %. North America follows with 16.9 %, South America with 15.5 %, Europe with 10.6 %, Africa with 9.6 %, and Oceania with 9.2 %. For the types of irrigation method, the CPIS proportion was highest in North America, with CPISs accounting for 13.8 % of the total irrigated area, followed by South America at 5.0 % and Oceania at 2.9 %.

In Fig. 8b, we summarize the irrigation and CPIS proportions across different climate zones. We used the global aridity index and criteria in the literature to classify climate zones (Zomer et al., 2022). The irrigation proportion decreases significantly, from 91.8 % in hyperarid zones to 20.7 % in semihumid zones. It then exhibits a slight increase to 21.4 % in humid zones. These variations in irrigation proportions correspond to the distinct water availability and climatic conditions in these regions. For the irrigation methods, the CPIS proportion is highest in the hyperarid region (5.7 %), followed by the semiarid region (3.9 %).

Figure 9a shows the irrigation proportion for each country. Notably, the irrigation proportion increases with geographical expansion from north Africa through west Asia, south Asia, and east Asia. In Fig. 9b, the irrigation proportions are presented for each IMZ. The spatial distribution aligns with the pattern depicted in Fig. 9a. Several countries in west Asia and north Africa, including Oman, Saudi Arabia, Qatar, and Egypt, boast irrigation proportions

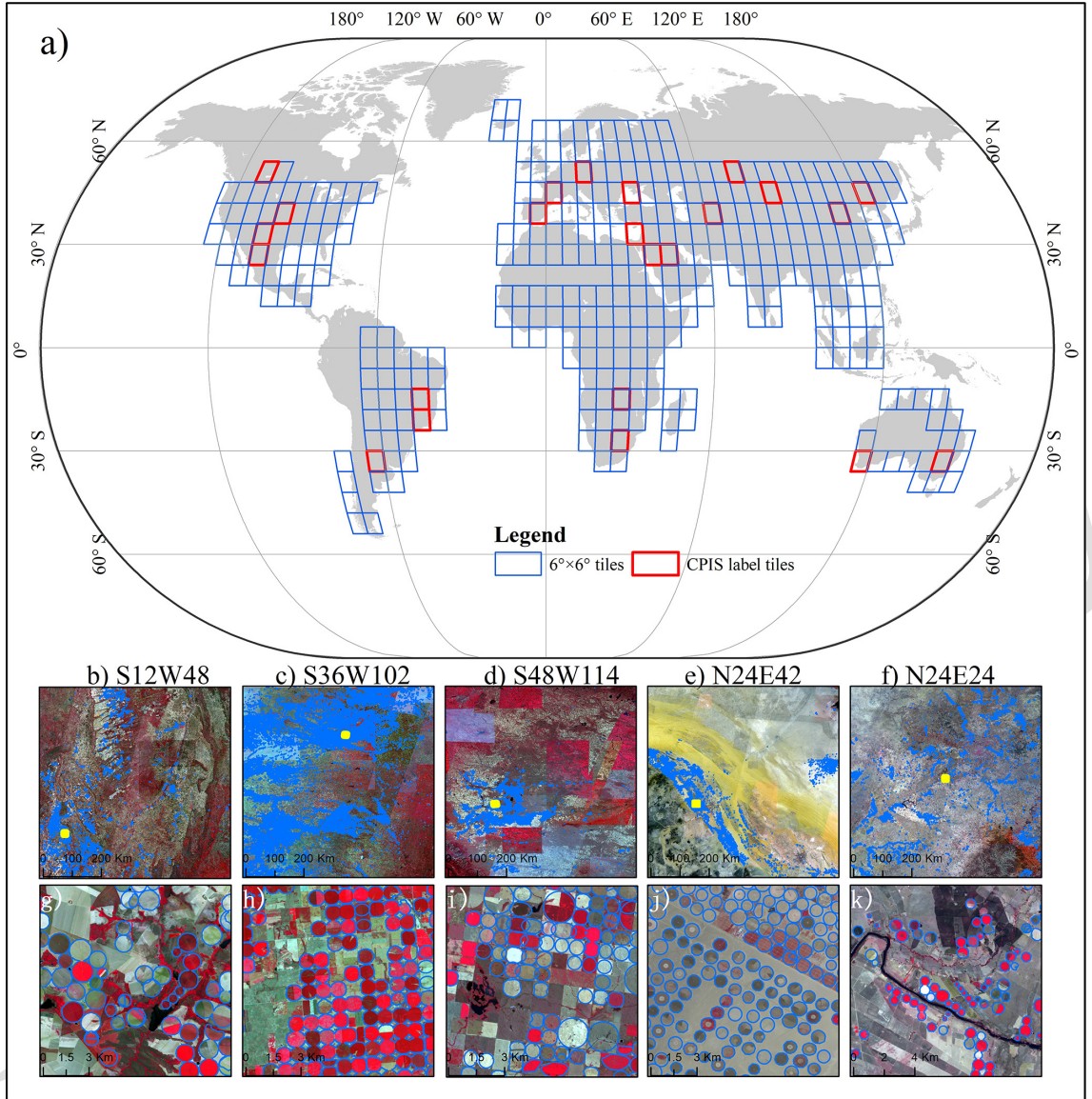

**Figure 5. (a)** Distribution of irrigation mapping zones and irrigated and rainfed cropland samples. **(b–f)** Five annotated tiles for CPIS labels and images. Panels **(b)–(f)** are the coordinates of the lower left corner point of each tile. Panels **(g)–(k)** are detailed maps of CPIS labels. Their locations are shown in panels **(b)–(f)** as yellow rectangles. The background images in panels **(b)–(k)** are Landsat-8 images.

of 100 %. Additionally, three countries surpassed an irrigation proportion of 80 %, namely, Turkmenistan (89.4 %), Uzbekistan (81.3 %), and Pakistan (80.4 %). Among all the IMZs, Gansu–Xinjiang in China has the highest irrigation proportion at 100.0 %, followed by the central northern Andes (96.2 %), Eurasian–African deserts (90.5 %), southern Himalayas in India (84.0 %), semi-arid Southern Cone (82.9 %), and Lower Yangtze in China (80.8 %).

Figures 9c and 8d are the CPIS proportions for each country and the IMZ, respectively. CPISs are mainly concentrated in countries with intensified agricultural regions and extreme arid zones, such as the Middle East. The highest proportion of CPISs is in Namibia (23.4 %), followed by the

United States (20.33 %), Saudi Arabia (16.3 %), South Africa (15.7 %), Canada (12.6 %), Zambia (12.5 %), the Gaza Strip (12.2 %), and Brazil (9.6 %). For the IMZs, the proportions of CPISs were greatest in the Amazon (C24) at 81.2 %, north of the High Plains (C12-4) at 42.5 %, south Zambia (C09-3) at 41.6 %, American northwestern Great Plains (C12-3) at 36.0 %, western Mongolia (C47) at 25.0 %, British Columbia to Colorado (C11) at 24.2 %, the American Cotton Belt to the Gulf Coastal Plain in Mexico (C14-1) at 22.8 %, and the southwest Mexican and northern Mexican highlands (C18) at 21.4 %.

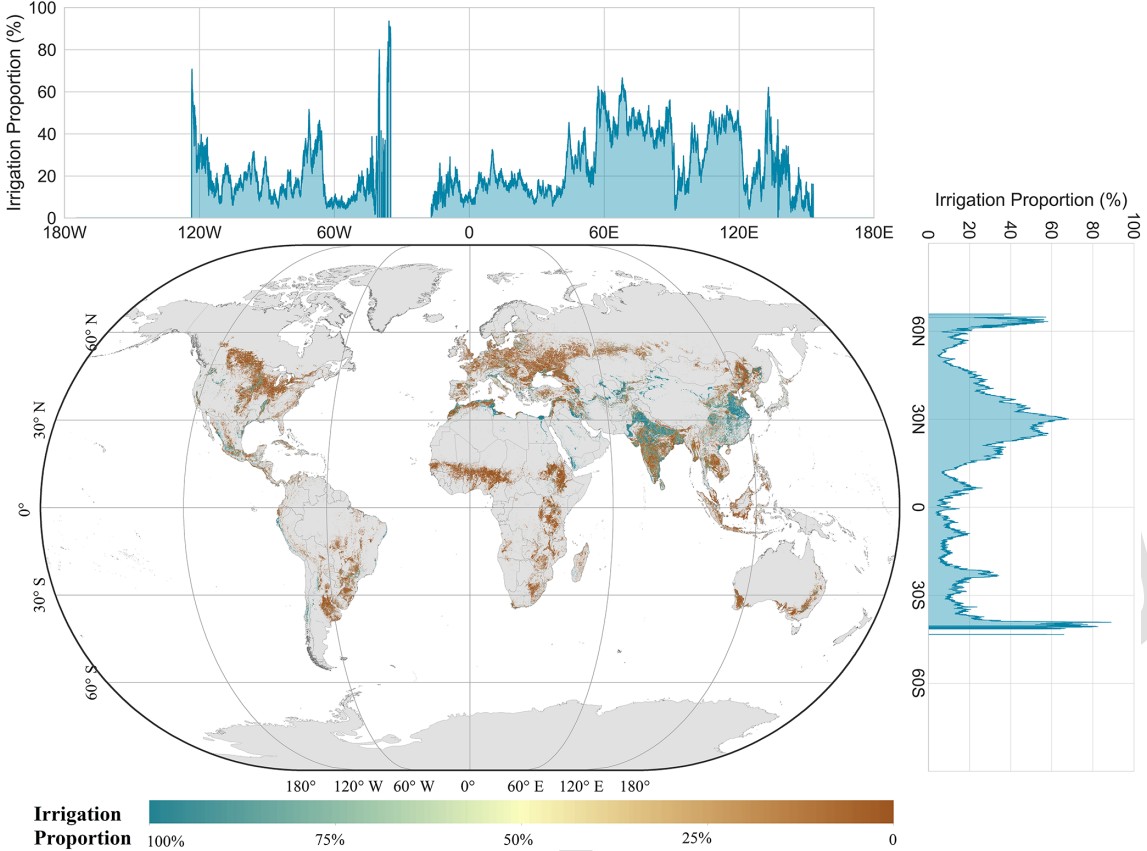

**Figure 6.** Global dataset of 100 m resolution irrigated cropland proportions.

## 3.2 Reliability of the GMIE-100

For each IMZ, the irrigation mapping method and threshold of the NDVI or $NDVI_{dev}$ are shown in Fig. 10. For the IMZ with a consistent dry season, the NDVI difference method was employed to determine the difference in amplification conditions between irrigated and rainfed cropland. To avoid the omission of fallow land and crop rotation, the maximum NDVI in the dry months of 2017–2019 was selected. The NDVI threshold for each IMZ was determined using training samples, which ranged from 0.10 in extremely arid regions, such as the Eurasian–African deserts (IMZ C64), to 0.74 in British Columbia to Colorado in North America (IMZ C11), as shown in orange in Fig. 10. These thresholds are integral to the accurate identification of irrigated cropland within each IMZ.

For regions without a significant dry season, the driest month of an extremely dry year among the 10 years (2010–2019) was selected to amplify the crop conditions between irrigated and rainfed cropland. The $NDVI_{dev}$ was used as a proxy to measure crop condition deviations from the 10-year average by using collected training samples. The values ranged from $-1.0\%$ (Amazon, C24) to $-37.0\%$ (C60-10, northwestern Greece and southwestern Albania), as shown in blue in Fig. 10.

Figure 11 shows the training accuracy of each IMZ. The $NDVI$ or $NDVI_{dev}$ threshold was determined using the Fisher discrimination method with 92 303 samples. Then, the training accuracy was assessed, which was between 0.31 % in the Amazon (C24) and 100 % in western Asia (C31-2). Despite the accuracy in some humid regions, such as northern South and Central America (42 %) and the Caribbean (49 %), there are 89 IMZs with accuracies greater than 80 % among the 105 IMZs with cropland. The specific accuracy for each IMZ is detailed in Table S1. The confusion matrix accuracy metrics of GMIE-100 are shown in Table 2. To validate the final accuracy of the GMIE-100, the remaining 20 % of the samples totalling 23 076 points were used. The overall accuracy of GMIE-100 was 83.6 %, with a user accuracy of 86.1 % and producer accuracy of 82.2 %.

The accuracy of GMIE-100 was evaluated in 10 countries, and the results are presented in Fig. 12, which shows the overall accuracy, user accuracy, and producer accuracy for each country. In China, the accuracy was assessed using 13 963 ground truth points derived from multiyear GVG data. The overall accuracy was 85.5 %, with a predicted accuracy of 86.7 % and user accuracy of 83.3 %. Commission and omission errors were prevalent in humid areas, such as southern China, Cambodia, and Myanmar. In other countries,

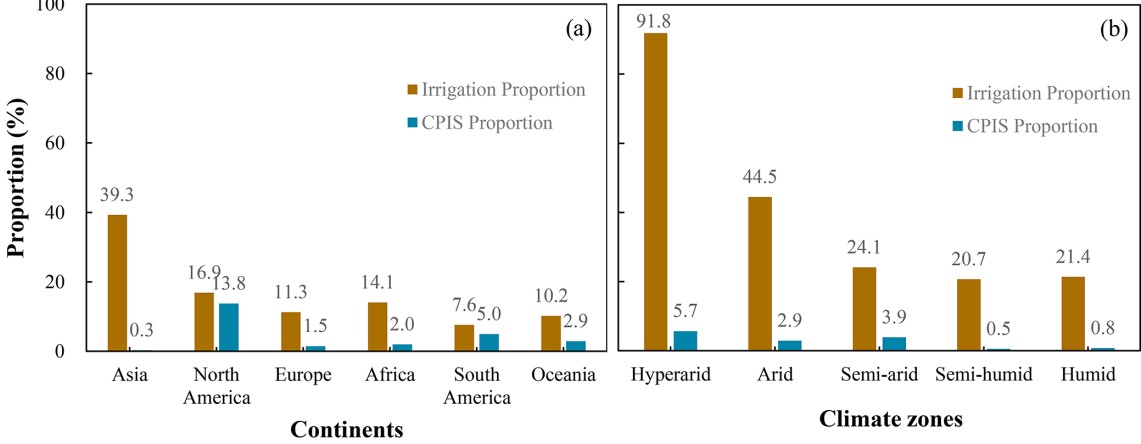

**Figure 7.** The distribution of irrigation types within the irrigation extent. Panels **(a)**–**(d)** show the detailed map of CPISs. The location of each subfigure is labelled in the main global map.

**Figure 8.** The irrigation proportion and CPIS proportion of total irrigated area for continents **(a)** and climate zones **(b)**.

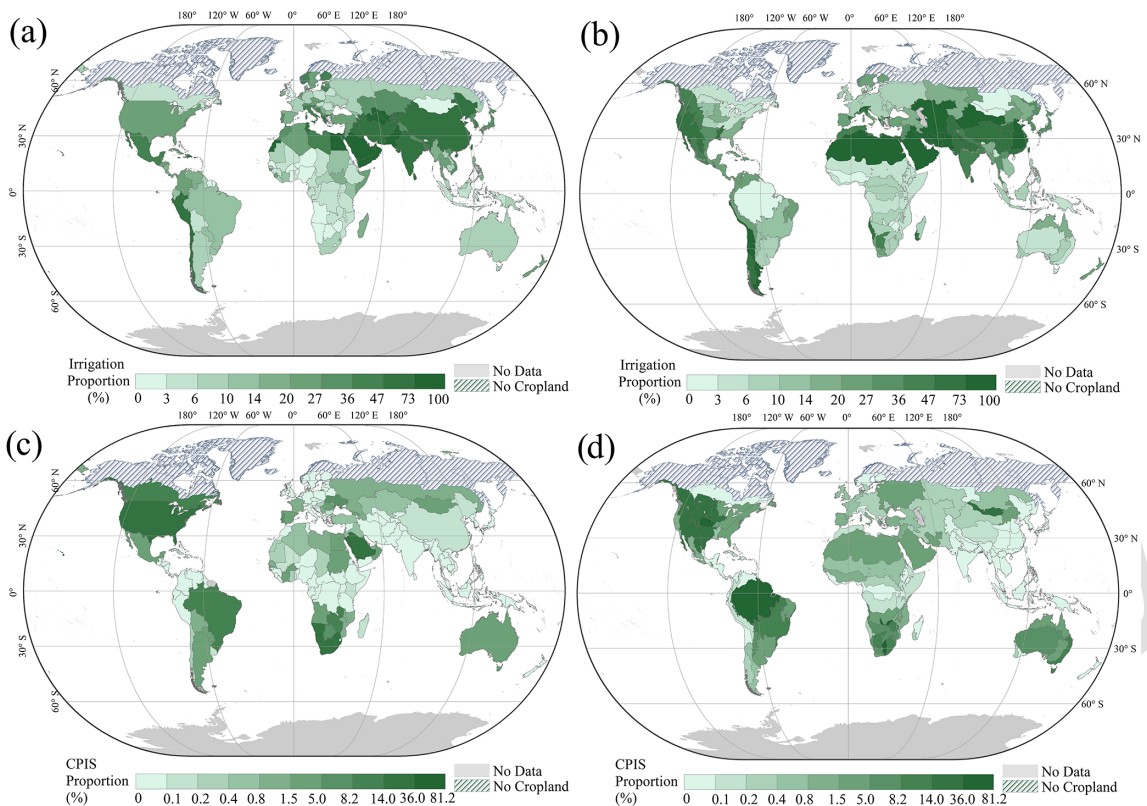

**Figure 9.** The irrigation proportion for each country **(a)** and IMZ **(b)** and the CPIS proportion of total irrigated cropland for each country **(c)** and IMZ **(d)**.

**Table 2.** Confusion matrix and accuracy assessment of GMIE-100.

|  |  | Field points | | | |
| --- | --- | --- | --- | --- | --- |
|  | Classes | Rainfed | Irrigation | Total | User accuracy |
| Predicted | Rainfed | 9270 | 2170 | 11 440 | 81.0 % |
|  | Irrigated | 1622 | 10 014 | 11 636 | 86.1 % |
|  | Total | 10 892 | 12 184 | 23 076 |  |
|  | Producer accuracy | 85.1 % | 82.2 % |  |  |
| Overall accuracy: |  | 83.6 % | | | |

the overall accuracy of the GMIE-100 datasets was generally acceptable.

The accuracy metrics and confusion metrics for the CPIS are listed in Table 3. The model achieved a high validation accuracy of 97.87 % ± 0.1 %. The $F_1$ score, which is a balance between precision and recall, is 86.87 % ± 0.1 %. The mean intersection over union (IOU) is 87.25 % ± 0.2 %. We visualized four patches with dense CPISs in Fig. 13. Overall, the CPIS was accurately identified in most cases.

## 3.3 Comparison with existing irrigation datasets

### 3.3.1 Comparison of irrigated cropland

To compare GMIE-100 against four existing irrigation products, we downscaled GMIE-100 and GRIPC-500 and USGS-LGRIP30 to a 1 km resolution and scaled IWMI-GIAM and FAO-GMIA to a 1 km resolution via the bilinear interpolation method. The results are shown in Fig. 14. The spatial pattern of irrigated cropland in GMIE-100 generally coincided with that of the other products. Irrigated cropland was most concentrated in the North China Plain and Ganges and Indus River basin from a worldwide perspective.

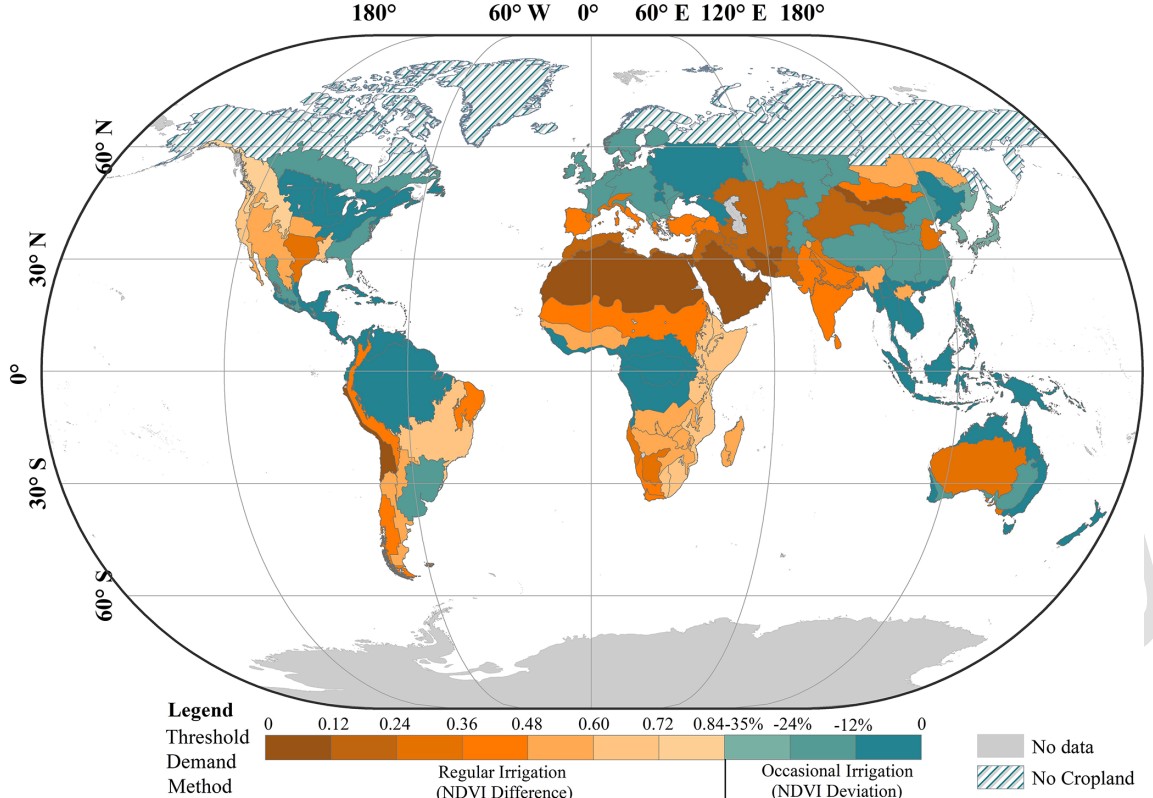

**Figure 10.** The thresholds of the NDVI difference and deviation for each IMZ.

**Table 3.** Confusion matrix of the GCPIS identified with Pivot-Net.

|              |   | CPIS predict | | Recall |
|--------------|---|-------------|-----------|-----------------|
|              |   | 0           | 1         |                 |
| CPIS label   | 0 | 119 938 874 | 735 300   |                 |
|              | 1 | 2 077 463   | 9 303 403 | 81.75 % ± 0.2 % |
| Precision    |   |             | 92.68 % ± 0.1 % | |
| Overall accuracy | |         | 97.87 % ± 0.1 % | |

Nevertheless, there were notable discrepancies in the detailed distributions of irrigated cropland patches, such as those in northeast China, the East European Plain, the Planície de la Plata of South America, and the lower Mississippi River basin (Fig. 15). In the North China Plain, the irrigated cropland appears denser within USGS-LGRIP30 and GRIPC-500 than in the other products. According to census data from China, the average irrigation proportion for three provinces (Heilongjiang, Jilin, Liaoning Province) is 39.32 %. According to the GMIE-100 results, the irrigation proportion is 27.45 %, which is closer to the census data. For the irrigated cropland in the East European Plain, USGS-LGRIP30 illustrates a broader distribution of irrigated cropland, which is significantly denser than that portrayed in GMIE-100 and the other three datasets (Fig. 15b1–b4). No-

tably, the GRIPC-500 dataset indicates a considerable extent of irrigated cropland in the Planície de la Plata region when compared to GMIE-100 and the other products (Fig. 15c1–c4). According to census data from Brazil, the reported irrigation proportion is 6 %, whereas it is 58 % and 72 % in USGS-LGRIP30 and GRIPC-500, respectively.

To validate the proposed GMIE-100, we compared it with national census data. The results are shown in Fig. 16. For comparison with existing global irrigation products, we also compared GMIE-100 against FAO-GIAM, IWMI-GMIA, USGS-LGRIP30, and GRIPC-500. The $R^2$ between the GMIE-100 and 23 national census datasets was 0.92, with an RMSE of 3.52 % and a MAE of 2.74 %. For FAO-GIAM and IWMI-GMIA, the $R^2$ values with GMIE-100 were 0.72 and 0.73, respectively. The determination coef-

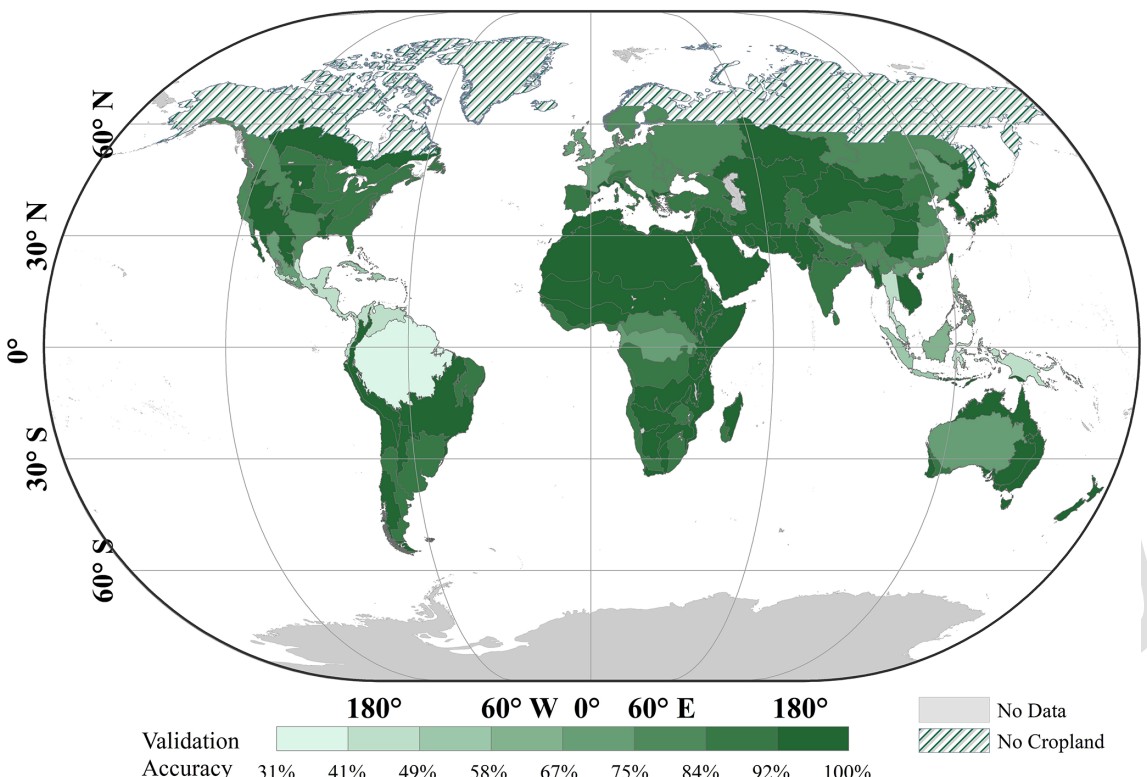

**Figure 11.** Training accuracy for each irrigation map zone.

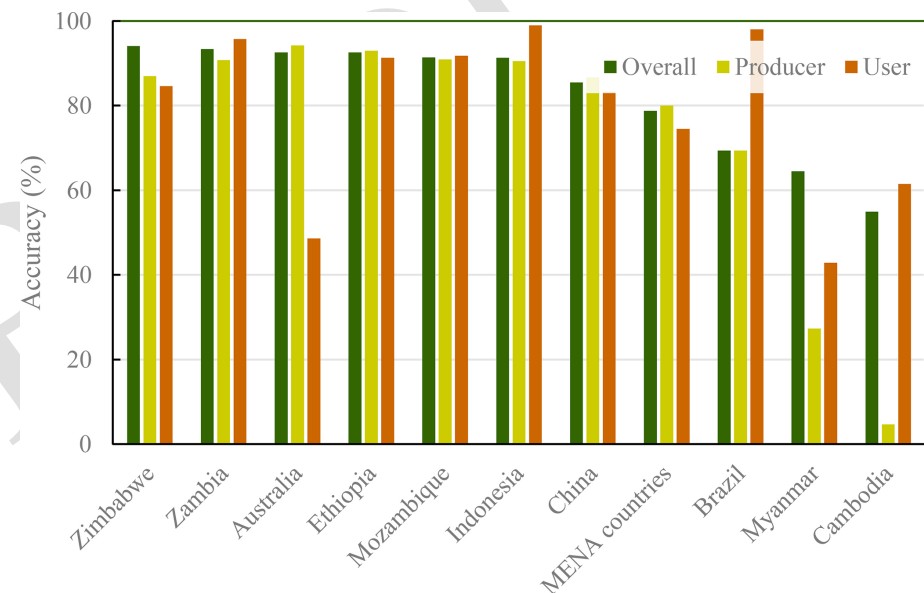

**Figure 12.** Accuracy for countries with GVG (GPS, Video, GIS) irrigation validation points.

ficient between USGS-LGRIP30 and GMIE-100 was only 0.45, with an RMSE of 35.6 %, the lowest value among these three existing irrigation products. When we compared USGS-LGRIP30 with the national census, the $R^2$ value was only 0.25. When comparing GMIE-100 with GRIPC-500,

the $R^2$ value was 0.51, with an RMSE of 29.89 %. The scatter plot shows that GRIPC-500 was consistently overestimated in comparison with GMIE-100.

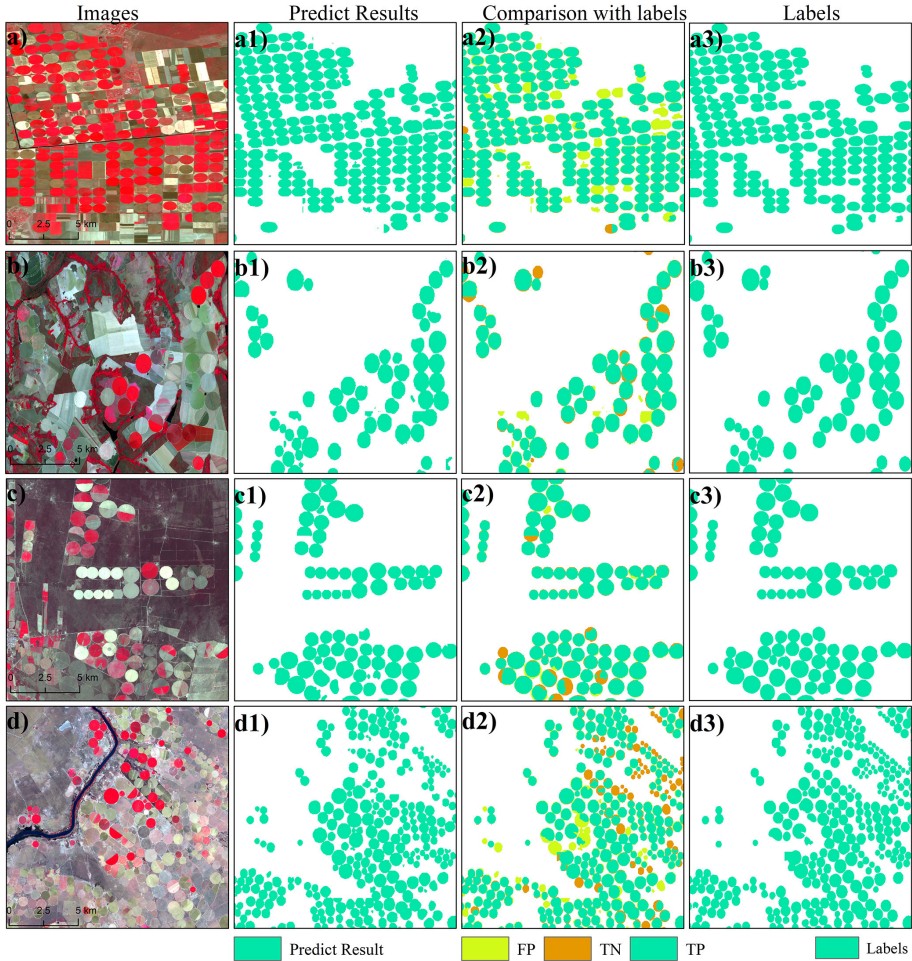

**Figure 13.** Accuracy assessment for the CPIS identification results. Panels **(a)**–**(d)** are the composted images; panels **(a1)**–**(d1)** are the prediction results of Pivot-Net, and panels **(a2)**–**(d2)** are the comparisons between our results and the labels. TP represents true positive pixels, while TN represents true negative pixels. FP means false positive samples. Panels **(a3)**–**(d3)** are the labels. The central point coordinates of panels **(a)**–**(d)** are (33.86, 46.37), (−47.34, 16.41), (−65.74, 32.03), and (25.11, 28.06), respectively. The background images in panels **(a)**–**(d)** are Landsat-8 images. Panels **(a)**–**(d)** are credited to the U.S. Geological Survey.

## 3.4  Advantages and limitations of GMIE-100

We used irrigation performance to map irrigation at regular intervals. Irrigation areas exhibit significant variability in irrigation water use (Puy et al., 2021, 2022). Thus, changes in the irrigated area could reflect variations in agricultural water use, which is important for local water resource management. Due to a lack of updated information, global maps of irrigated areas often rely on estimates from approximately 2000 (Nagaraj et al., 2021). For RIR regions, irrigation maps can be updated every 3 years by collecting the vegetation signals during each dry season. For RIO regions, irrigation maps can be updated every 10 years based on crop status during extremely dry events. Although the irrigated cropland extent during the dry season can be identified from 2010 to 2019, our aim was to provide the most up-to-date information based on satellite data over the 2017–2019 period.

Periodic cropland fallowing refers to the practice of not cultivating or tilling all croplands within a single year. This approach is often employed to restore soil fertility as part of a crop rotation scheme or to prevent excess agricultural production. The use of the NDVI or $NDVI_{dev}$ thresholds enables the identification of only those lands that have been actively cultivated. Subsequently, these cultivated lands can be further categorized into either irrigated or rainfed land. An area is designated as irrigated if it has been cultivated at least once during the driest month over a span of 3 years. This criterion aids in discerning areas that are actively managed for crop production from those temporarily left fallow or unplanted.

The spatial resolution of this dataset was 100 m, which is greater than that of the dominant irrigation data map. High-resolution irrigated cropland data are essential for quantifying agricultural water use (Wu et al., 2022). The resolution of most existing irrigation data is very coarse, varying between

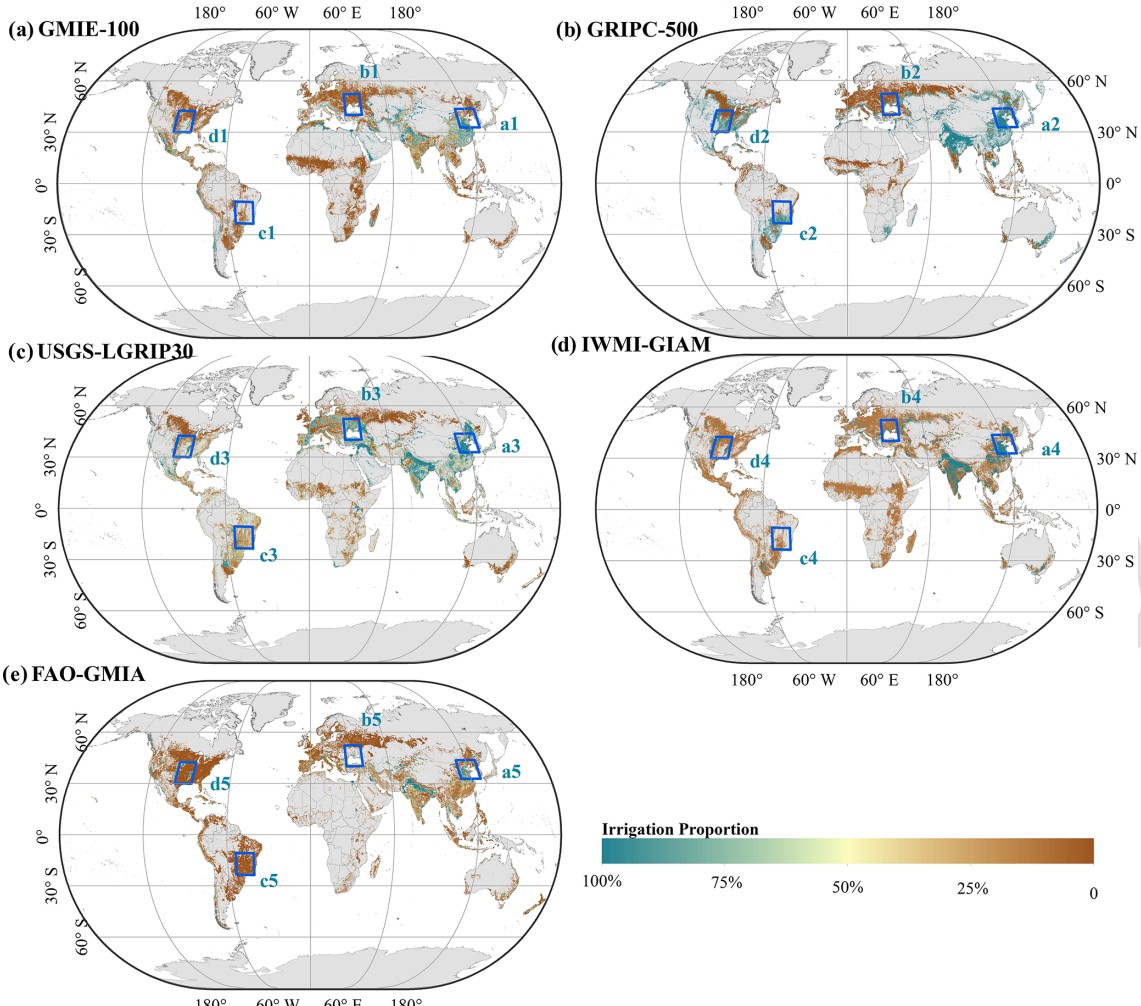

**Figure 14.** Comparison of existing irrigation production at 1 km (GMIE-100, GRIPC-500, USGS-LGRIPC30) or 10 km resolution (IWMI-GIAM and FAO-GMIA).

500 m and 10 km (Xie et al., 2019). As shown in Fig. 17, GRIPC-500, IWMI-GIAM, and FAO-GMIA fail to capture detailed information on irrigated cropland. Even though the resolution of USGS-LGRIPC-30 was greater than that of
5 GMIE-100, the latter descriptions of heterogeneous irrigated cropland distributions in the North China Plain (Fig. 17a1 and a3) and the US Plateau (Fig. 17d1 and d3) were better than the former. An evapotranspiration–precipitation product with 500 m resolution was used to determine the driest
10 months within each IMZ, and the time period was used to detect irrigation performance and detect irrigated cropland. Within each IMZ, 30 m NDVI data were used as major input. Then, to avoid effect fallow land and crop rotation, we calculate the irrigation proportion within 100 m.
15    As for the maximum extent, it should be understood separately for RIR and RIO. For RIR, the largest area refers to the cropland area irrigated at least once over the past 3 years (2017–2019) because we detect irrigation every year for this

region. To avoid missing fallow land, we identify the largest extent over the 3-year period (2017–2019). For RIO, it means 20 the cropland area was irrigated at least once in the last decade (2010–2019). For RIO, irrigation occurs occasionally. We determine whether the cropland is irrigated in the driest year. But in the normal year, the irrigation may not be necessary in this area. Thus, we identified the largest extent area for the 25 last 10 years (2010–2019). On the other hand, when we compare our result with nation census data, the result shows high consistency. Compared with USGS-LGRIP30 and GRIPC-500, our result does not exhibit significant overestimation.

   When discussing irrigation extents, it is crucial to differen- 30 tiate between "net irrigated area" and "gross irrigated cropland area". The net irrigated area refers to the actual land area equipped with irrigation facilities and receiving irrigation, while the gross irrigated cropland area encompasses all the land that can potentially be irrigated during a crop's 35 growing season, regardless of whether it is continuously irri-

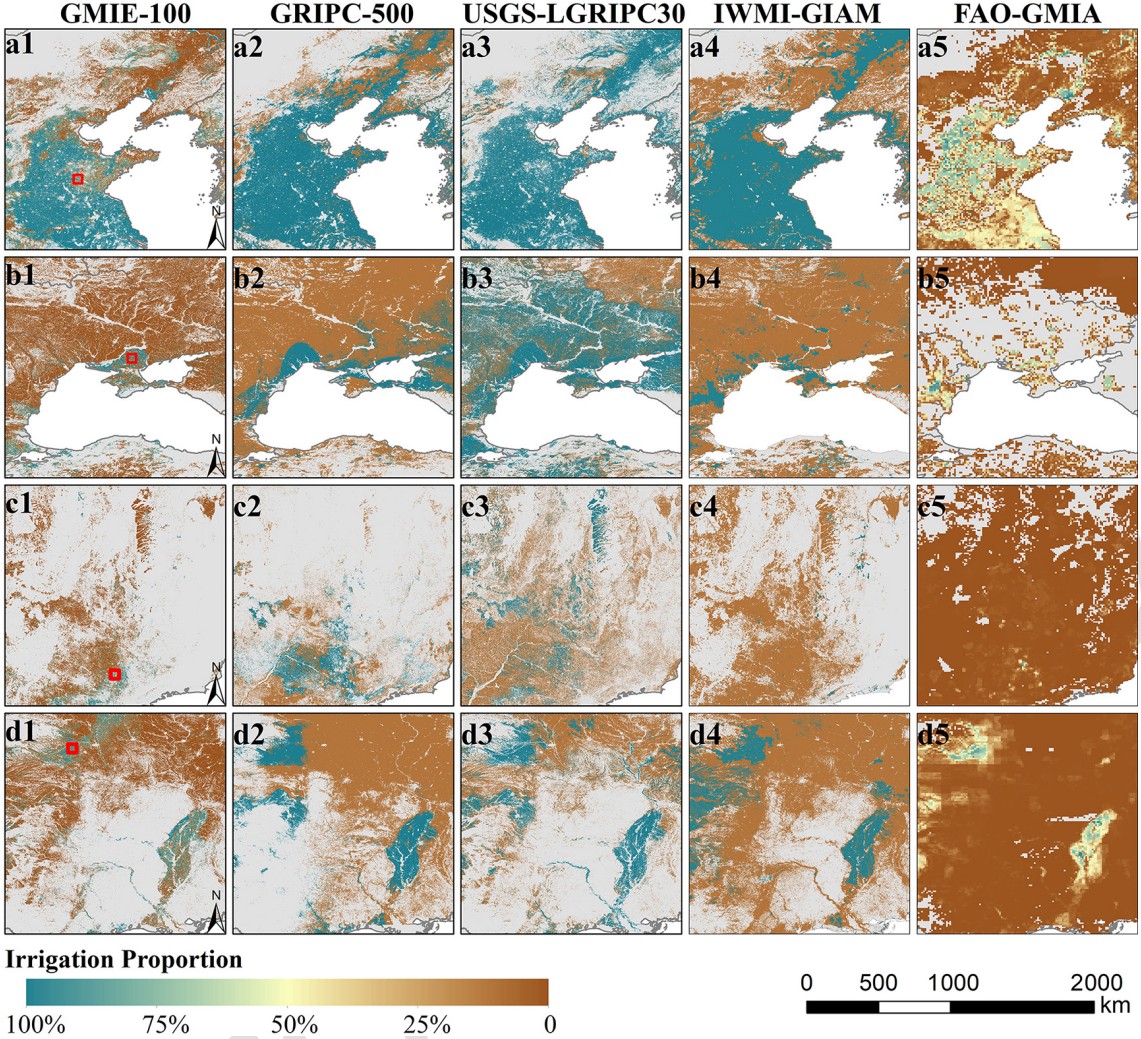

**Figure 15.** Comparison with existing irrigation production for the hotspot region of irrigation. The corresponding location is labelled in Fig. 14 with a blue rectangle.

gated throughout the season. For instance, if a plot of land is planted and irrigated twice in one growing season, that land should be counted twice, reflecting in the gross irrigated cropland area. Therefore, the gross irrigated area typically exceeds the net irrigated area because it accounts for instances of multiple plantings and irrigations. This distinction is vital for accurately assessing the use of water resources and planning agricultural production. In our research, we estimated maximum irrigation extent under the assumption that irrigation equipment is specifically deployed to mitigate the most water-stressed conditions. Thus, we estimated the net irrigation area for the selected growing season, whose value should be largest during that decade or 3-year period. For RIR, we estimate the net irrigation in the dry season and growing season that experiences the greatest water stress each year. Similarly, for RIO, we evaluate net irrigation area

based on a single growing season that has undergone an extreme drought event in the last decade.

Furthermore, with the support of the DL method, we successfully mapped CPISs globally, which enabled our investigation of specific irrigation methods. We found $11.5 \pm 0.1$ Mha of CPISs worldwide, composing $2.90\% \pm 0.03\%$ of the total irrigated cropland. To the best of our knowledge, this is the first study to map the CPIS irrigated method, despite Chen's research on CPI mapping in global arid regions (Chen et al., 2023a). GMIE comprises both the irrigated cropland extent and specific irrigation method (CPIS) distributions with relatively high resolution, thus providing subbasin water consumption and withdrawal estimations for all sectors (Wu et al., 2022). Due to the variation in irrigation efficiency among different types of irrigation methods, CPISs demonstrate an efficiency exceeding 80 %, while gravity-flowing irrigation methods ex-

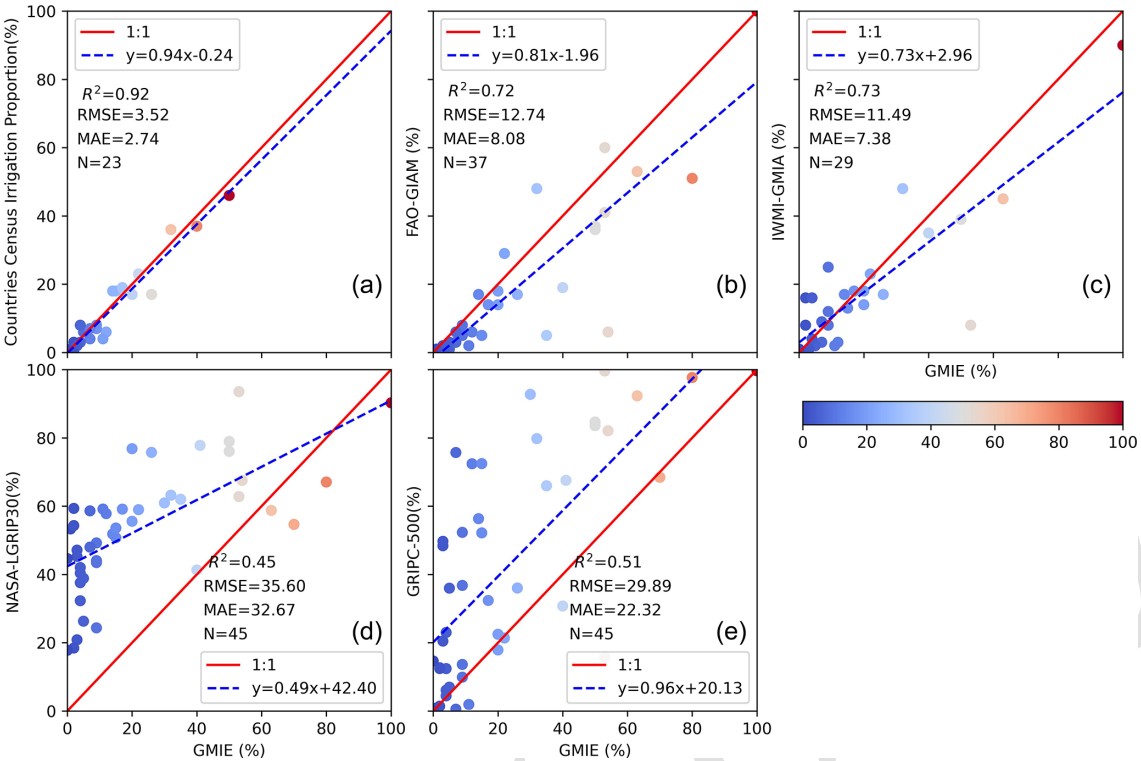

**Figure 16.** Comparison of national irrigation proportions between GMIE-100 and national census data **(a)**, FAO-GMIA **(b)**, IWMI-GIAM **(c)**, USGS-LGRIP30 **(d)** and GRIPC-500 **(e)**.

hibit a comparatively low efficiency of approximately 60 % (Waller and Yitayew, 2016). Therefore, irrigation efficiency can be estimated based on types of irrigation methods in the future. This process could enhance the understanding of the irrigation paradox (Grafton et al., 2018), which indicates that technological advancement increases irrigation efficiency, but crop water levels do not decrease. However, this study excluded other irrigation types because the identification of CPISs relied on the circle shape in the satellite data, and other irrigation types lack this distinct feature. The identification of other irrigation types in the future will be important for water use estimations (Boutsioukis and Arias-Moliz, 2022), maybe with the help of big geodata. In the maximum irrigation extent, we include all the irrigation types that could mitigate water stress.

Compared to the surveillance classification method, our method requires fewer samples. However, due to a lack of expertise, all spectral characteristics of irrigated farmland were studied using training samples, which increased the required number of samples. Xie's research used 20 000 samples for irrigation mapping in the United States (Xie et al., 2019). Zhang's research used approximately 100 000 samples to identify irrigated croplands in China (Zhang et al., 2022b). By determining the NDVI difference and NDVI deviation between irrigated and rainfed cropland, the required number of training samples could be drastically reduced. In

this study, a total of 92 303 samples were used to determine the NDVI threshold and the NDVI deviation threshold at the global scale. Moreover, training samples in China were mostly collected on site, which is more precise than visual interpretation.

Additionally, there are several limitations to this method. Firstly, the accuracy of our method varies significantly across different regions due to the variability in climate, soil types, crop species, and irrigation practices among different areas. According to the accuracy reports for each irrigation mapping zone, cropland is present in 105 out of the total 110 irrigation mapping zones, with 96 of them exhibiting an accuracy greater than 70 %. There are only nine divisions with accuracies below 70 %, most of which are situated in the southeast Asian island countries, such as Thailand, Myanmar, and Laos, and the tropical rainforest areas of South America, notably the Amazon, characterized by their humid conditions. We acknowledge that there is significant uncertainty in these aforementioned regions; however, the proportion of irrigation in these areas is typically not as substantial compared to arid and semi-arid regions. The task of identifying irrigation in these regions using machine learning methods is also challenging, as it is not straightforward to fully distinguish between irrigated and rainfed cropland without accurate phenological inputs. A potential solution for improving accuracy in humid regions could involve the integration of ir-

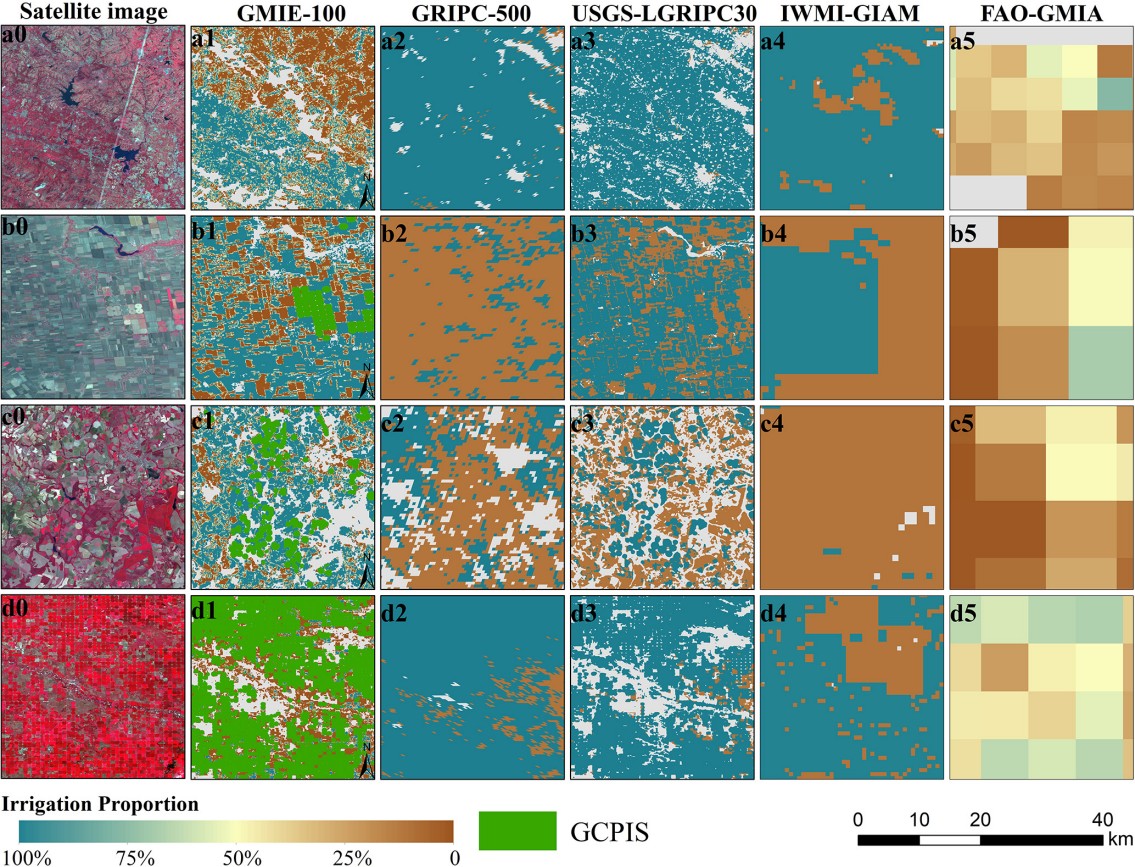

**Figure 17.** Comparison between GMIE-100 and existing global irrigation products in detail; their specific locations are labelled in the corresponding subfigure Fig. 15 with red rectangles.

rigation performance assessments to select optimal time windows, coupled with advanced machine learning techniques. Additionally, the representativeness of sample points can be further improved, e.g. by identifying CPISs via DL methods (Tian et al., 2023b; Chen et al., 2023a), which are commonly used in the United States, Brazil, and the Middle East.

Second, although GMIE-100 provides a relatively high-resolution distribution of irrigated cropland, it produces some mixed pixels with cropland or noncropland and irrigated or rainfed cropland. This is especially true for regions with extremely small agricultural fields (Fritz et al., 2015). The cropland masks had the greatest influence on the GMIE-100 dataset (Salmon et al., 2015; Meier et al., 2018), despite the selection of 16 distinct cropland datasets derived from country- and region-level sources as high-priority inputs. These datasets often demonstrate disparities in estimating the distribution of cropland, particularly in African countries, due to the complex landscape, frequent cloud cover, and the presence of small agricultural fields (Nabil et al., 2020). Consequently, inaccuracies within the cropland datasets transferred onto the GMIE-100 dataset. Nevertheless, importantly, these datasets remain the primary sources of cost-effective and up-to-date information covering vast geographical areas.

Actually, we just focus on seasonal cropland because permanent crops are generally used for fruit trees, nut trees, coffee, tea, and some types of vines; this is recognized as shrub or tree in most land cover systems such as ESRI (Karra et al., 2021), FROM-GLC (Yu et al., 2013), GLAD_Map (Potapov et al., 2022), GLC-FCS30 (Zhang et al., 2021b), and WorldCover (Zanaga et al., 2022). Conversely, harvest crops, maize, soybean, wheat, and rice are most important for food security. Therefore, we choose this definition to distinguish irrigated and rainfed cropland rather than the definition from FAO. Different crop definitions as input data may produce varied irrigated cropland area, which will definitely introduce uncertainty in the final result. A consistent, high-resolution cropland mask with high accuracy is urgently needed to solve this problem.

Thirdly, because it is challenging to collect the global field samples, we fused three sources of samples. From different country, there are varied dominant sample sources. For instance, in China, most samples were collected from the GVG field survey, while in Brazil, major samples were from the USGS. Except countries with GVG and USGS samples, the visual interpretation data were the dominant source of samples. This also ensures the represented manner of irrigated

cropland. Overall, the number of samples was very large. Basically, this irrigated and rainfed sample database suffices for the globally irrigated cropland mapping compared with global cropland expansion mapping research (Potapov et al., 2022), which achieved cropland mapping globally with thousands of samples. Meanwhile, these fused samples maybe introduce some uncertainty in terms of representation. This effect should be acceptable in arid and semi-arid regions because the irrigation performance is relatively easy to identify. However, the uncertainty may increase in wet regions due to the complexity of irrigated cropland. Also, a parcel of land is designated as irrigated if it receives any supplemental artificial water supply to support crop cultivation at least once during the growing season. The Global Maximum Irrigated Extent (GMIE) dataset, initially developed at a 30 m resolution, categorizes each pixel as either irrigated or rainfed cropland. Thus, even if a pixel contains less than 100 % irrigated cropland, it is classified as an irrigated pixel within that $30 \times 30$ m area. As a result, there may be a tendency towards overestimation due to the mixed pixels at the 30 m resolution, particularly in regions with smaller fields such as southern China, southeast Asia, and parts of Africa. However, the relatively high resolution of the pixels helps to mitigate this uncertainty to a certain extent.

## 4   Code and data availability

The data are publicly accessible through the following link: https://doi.org/10.7910/DVN/HKBAQQ (Tian et al., 2023a). The GMIE-100 dataset spans values ranging from 0 to 1, with a designated no-data value of $-99$. Globally, there are 67 tiles available, each with a maximum extent of $21° \times 21°$. In cases where these tiles overlap with land, they maintain the standard extents; however, adjustments are made to the tile extents as needed to accommodate the terrestrial range. The GCPIS was stored in shapefile format in zip files. The irrigation unit zone can be downloaded from https://doi.org/10.7910/DVN/HKBAQQ (Tian et al., 2023a). TS4

## 5   Conclusion

High-resolution and updated irrigation maps are important for tracking regional water use and food production situations. Using irrigation performance data collected during the dry season of the growing season and during extreme drought events, we produced the GMIE-100 with a 100 m resolution with the support of GEE. In this study, the entire globe was divided into 110 zones based on variations in climate and phenology. In each IMZ, we identified the dry months during the growing seasons from 2017–2019 or, alternatively, the driest months during the most arid year from 2010 to 2019. To distinguish irrigated cropland, we employed 92 303 samples to determine thresholds for the NDVI during the dry months of 2017 to 2019 and the NDVI deviation

from the 10-year average for the driest month ($NDVI_{dev}$). The NDVI or $NDVI_{dev}$ threshold that achieved the highest overall accuracy was selected to distinguish irrigated and rainfed cropland. All the algorithms were conducted using GEE with the code https://code.earthengine.google.com/eaafaab35dde9bbe37f443e80c716479 TS5 (last access: 17 February 2025).

With the support of the DL method, the global CPIS was identified using Pivot-Net. We identified 11.5 Mha of CPIS irrigated cropland, accounting for approximately $2.90 \% \pm 0.03 \%$ of the total irrigated cropland. However, in Namibia, the United States, Saudi Arabia, South Africa, Canada, and Zambia, the proportion of CPISs was greater than 10 %. To our knowledge, this is the first attempt to identify types of irrigation methods globally, although other types of irrigation methods, such as gravity flow, are still dominant types of irrigation methods. Nevertheless, our approach facilitates the estimation of irrigation efficiency based on different types of irrigation method proportions to support high-accuracy subbasin-scale water resource management.

Finally, the GMIE-100 was produced with a 100 m resolution. Using 23 076 points to validate the results, we found that the overall accuracy of GMIE-100 was $83.6 \% \pm 0.6 \%$, but it varied among the different IMZs. The GMIE-100 indicates that the largest extent of irrigated cropland reached $403.17 \pm 9.82$ Mha, which accounts for $23.4 \% \pm 0.6 \%$ of the total global cropland. Spatially, irrigated cropland is concentrated in great plain regions and regions near rivers. A total of 224 Mha of irrigated cropland, accounting for 55.6 % of the total irrigated cropland, is in the plains regions. The Ganges Plain, the Indus Plain, and the North China Plain all have large amounts of irrigated cropland worldwide. The GMIE-100 provides more detailed information about irrigated and rainfed cropland and thereby improves its ability to support agricultural water use estimation and regional food situation assessment.

**Supplement.** The supplement related to this article is available online at [the link will be implemented upon publication].

**Author contributions.** HZ and BW conceptualized the study. FT designed the experiments and carried out the experiments. BW and HZ were responsible for funding acquisition. MZ and WZ conducted the investigation and formal analysis. FT prepared the original draft of the manuscript. FT, BW, HZ, MZ, WZ, NY, YuL, and YiL reviewed and edited the manuscript.

**Competing interests.** The contact author has declared that none of the authors has any competing interests.

**Disclaimer.** Publisher's note: Copernicus Publications remains neutral with regard to jurisdictional claims made in the text, pub-

lished maps, institutional affiliations, or any other geographical representation in this paper. While Copernicus Publications makes every effort to include appropriate place names, the final responsibility lies with the authors.

**Acknowledgements.** We gratefully acknowledge the support of the Google Earth Engine platform, which provided essential computational and storage resources, simplifying access to archived datasets such as TM/ETM/OLI satellite data, TRMM and GLDAS for precipitation data, and MOD16A2.006 for evapotranspiration data. These resources greatly facilitated programme calculations and data retrieval. We thank the data provider for the abovementioned data and the GFSAD30 team for publishing the irrigated and rainfed samples. Furthermore, we would like to express our gratitude to the authors of existing irrigation datasets, namely, GRIPC-500, USGS-LGRIP30, IWMI-GIAM, and FAO-GMIA, for their foundational work, which has significantly contributed to our research in this field. Their efforts have provided essential background information for our study.

**Financial support.** This research was supported by the Second Tibetan Plateau Scientific Expedition and Report (TS6 grant no. 2019QZKK0308), the National Natural Science Foundation of China (grant nos. 2042301409, 41861144019, 42071271), the Chinese Academy of Sciences–Max Planck Society (CAS-MPG) research project (grant no. HZXM20225001MI), the strategic consulting project of the Alliance of International Science Organizations (grant no. ANSOSBA-2022-02), and the National Key Research and Development Program of China (grant no. 2016YFA0600304).

**Review statement.** This paper was edited by Kaiguang Zhao and reviewed by four anonymous referees.

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

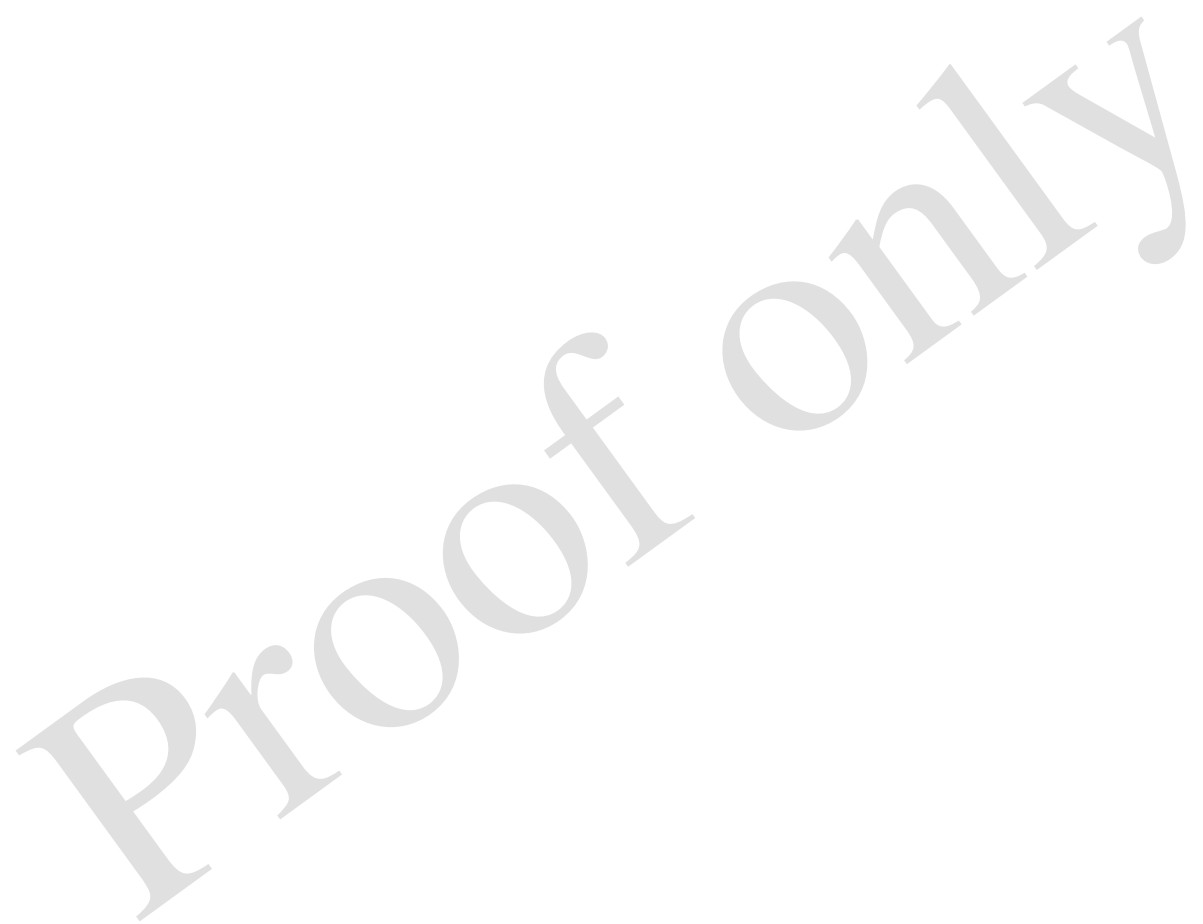

## Remarks from the typesetter

TS1    Please confirm adjusted order of the reference.

TS2    Please give an explanation of why this needs to be changed. We have to ask the handling editor for approval. Thanks.

TS3    Please confirm added citation

TS4    Please confirm this section.

TS5    Please check URL.

TS6    Please confirm added "grant no.".

TS7    Please provide location and date of the conference, as well as DOI/URL.

TS8    Please provide URL/DOI.

TS9    Please confirm the title.