# Peer review of "GMIE-100: A global maximum irrigation extent and central pivot irrigation system dataset derived via irrigation performance during drought stress and machine learning methods"

_Earth System Science Data, 2023_

## Referee Comment (RC1)

Review of: "GMIE-100: a global maximum irrigation extent and irrigation type dataset derived through irrigation performance during drought stress and machine learning method."

essd-2023-536

**General comments**

This article describes a dataset purporting to describe maximum irrigation extent and irrigation type with global scope at a 100-metre resolution. This dataset would have broad applicability for agricultural, economic and other analyses at global and more localised levels.

The authors make an attempt at providing this dataset at such a refined resolution, however there are some fundamental issues that need to be addressed before it could actually deliver what the authors promise in the article. I believe currently the authors give a flawed sense of accuracy in their estimates of irrigated and non-irrigated land. In its current form I do not recommend this manuscript/dataset be accepted for publication in ESSD.

**Major comments**

1. **Areas and cropland definition:** This dataset/manuscript needs better clarification of what areas of irrigated and non-irrigated land are included. For instance the title suggests the dataset is global, implying all irrigated and non-irrigated land are included. In the abstract they state 'In our study, we present a robust methodology that leverages irrigation performance during drought stress as an indicator of crop productivity and water consumption to identify global irrigated _cropland._' The latter implies it includes only cropland. Cropland has different definitions to different authors (see Tubiello et al 2023: https://www.nature.com/articles/s43016-022-00667-9) and can be very tricky to differentiate properly. In section 2.3 the authors state they use the JECAM definition of cropland which includes land used for seasonal crops (sowed/planted and harvested at least once within the 12 months) such as cereals, root and tuber crops, oil crops as well as economically significant crops like sugar, vegetables, and cotton. Additionally land occupied by greenhouses was considered as cropland. Greenhouses in cropland is a strange inclusion and needs explaining. The authors then go on to say they used "The cropland mask at 30-meter resolution could be obtained from International Research Center of Big Data for Sustainable Development goals via https://data.casearth.cn/thematic/cbas_2022/158". They state the overall accuracy of this dataset is 89.4%, but when I look at maps from these data it appears as though they include a lot of non-cropland area esp. pasture and meadow land (see Fig 1 below). I therefore do not have confidence that this dataset is suitable for supporting the authors assertion that their dataset has 100 metre resolution. Furthermore, the title of this manuscript implies this dataset is for 'maximum irrigation extent' i.e. all irrigation. They assess centre pivot irrigation, but it is not clear if the authors include lateral irrigators which is much the same technology as centre pivot, only it could be harder to distinguish lateral irrigation due to the patterns of NDVI (see figure 12). Finally, as per section 2.1 the research relied on evapotranspiration data at a 500 m resolution. Shouldn't the authors state that the resolution of their irrigation dataset is equivalent to the lowest

resolution of their input data? Otherwise you are giving a false sense of accuracy.

[Figure]

*Figure 1: Screenshot of casearth.cn datset.*

2. Given the above uncertainties in cropland categorisation I suggest the authors use a definition of cropland that aligns to something like that used by the FAO. This will improve the applicability of the dataset.

3. This manuscript needs to be edited heavily before it is resubmitted. I made a note of some of these edits in minor comments in the first few pages. Note, the list I provide is not exhaustive as there were many other changes to make.

4. Lines 134-140. A better plain language description of how irrigated and non-irrigated land was categorised is needed.

5. Section 3.4. The uncertainty in estimates of cropland used in the authors models needs to be better explained. Differences in classification of 'cropland' for instance can contribute to variation in estimates in irrigated cropland mentioned in section 3.1.

**Minor comments**

1. Abstract 1st line 11. "primary sector of human water…"; Use other word than sector such as form.
2. Line 26: What is the DL method? Define when you first use an abbreviation.
3. Line 27: What is Pivot-Net?
4. Line 29: "The GMIE-100 dataset containing both or irrigated extent…". What does the both relate to?
5. Line 40 use reference to back up claim that highest resolution maps are 500m to 10km.
6. Line 60 use space between croplands and (Thenkabail et al 2009)
7. Line 106. Use reference to back up claim of 80% efficiency.
8. Throughout references and tables, make sure abbreviations are defined esp in title of Figure 1 and 2.
9. Line 175. What is GVG?
10. Line 251. Spelling mistakes in Nirrgated and Nnon-irrgated.
11. Line 268. Spelling mistake exemple.
12. Line 377: belt_Mexican coastal plain. Error.

13. Line 469-471: How does looking at if an area of land has been cultivated during the driest month over a span of three year help determine if it is irrigated land? What if the cultivation occurs in one of the regular wet seasons of the year but irrigated is still needed thereafter?

---

## Referee Comment (RC3)

[referee-annotated manuscript omitted]

---

## Author Comment (AC1)

We are highly appreciated for your constructive comments and suggestions on our manuscript. Those comments and suggestions are valuable and helpful for revising and improving our article, as well as inspiring our research. We have carefully reviewed the comments and have revised the manuscript accordingly. Our responses are given in a point-by-point manner below and BLUE fonts. Please find our detailed responses below to all these comments/suggestions and thank you again for everything you have contributed.

**RC2**

This study demonstrated a global irrigation dataset with 100 meters using irrigation performance during drought stress, which is a brand-new way to detect irrigated and non-irrigated cropland. Furthermore, this MS finishes mapping the global central pivot irrigation system using the Deep Learning method. Also, it is interesting to detect special irrigation methods using deep learning methods. Overall, the MS was well-written and designed for readers. But there were still some concerns before this MS was accepted:

Response: Thanks for your positive comments.

Major concerns:

1. About the resolution: In section 2.1 some coarse data was described as input data, but the final resolution of the irrigation map is 100 meters, so this will mislead some readers on how to produce a 100-meter irrigation map using 500-meter data.

   Response: Thanks for your comments. The evapotranspiration, precipitation product with 500-meter resolution was used to determine the driest months within each IMZ. And the time period was used to detect irrigation performance and detect irrigated cropland. In each IMZ, 30meter NDVI data was used as major input. Then to avoid effect fallow land and crop rotation, we calculate the irrigation proportion within 100meters.

   We also added this statement in the body text.

2. About the IMZs: You mention that "65 MRUS in Cropwatch served as the basis for further division of global cropland into 110 irrigation mapping zones (IMZs)", what is the principle for further dividing these zones? Are these zones available or not?

   Response: Thanks for your comments. We further divided 65 zones into 110 based on arid indices, water availability, soil types, and landforms. This data is publicly available on the CropWatch website or you can contact us via email.

3. About accuracy assessment: you collect many field points using the GVG app. How to distinguish irrigation field points during the field survey?

Response: Thanks for your constructive suggestions. Although it is not easy to identify irrigated cropland on satellite data, irrigation cropland could be identified accurate in field according to irrigation infrastructure, crop type and crop health condition. Even you cannot distinguish them following above characteristics, you could ask local farmer, who will answer this question with hesitate.

- Irrigation infrastructure, some obvious feature was easy to identify, such as canner, irrigation plump and central pivot irrigation system. We display serval photos for this case as below:

[Figure]

| | |
|---|---|
| Irrigation cannel in Xinjiang | Drip irrigation in Hebei province |
| Irrigation pump | Central pivot irrigation system |

- Usually, irrigated was applied for certain crop types, such as winter wheat in North China Plain, Cotton in Xinjiang and vegetable and tomatoes in most province, et.al.
- Last but not least, irrigated crops usually appear greener and lush compared with near crops.

4. The irrigation map and GCPIS were identified using two ways (irrigation performance and DL), but some figures make me confused to display these two results.

   Response: Thanks for your valuable comments, we have changed the display manner in the MS for Figure1, 6, 16.

Minor revision:

1. The preprocess of NDVI data in Line 160 should be further explained.

   Response: Thanks for your suggestion. We added more explanations in the text to describe the preprocessing to NDVI data.

2. You could list some detailed maps of global CPIS in Figure 6 to make the global CPIS map clearer.

   Response: Thanks for your suggestion, we added the detail map of CPIS in Figure6.

3. In Figure 16 it will be significant if the satellite images were added to give the reader a basis for their judgment.

   Response: Thanks for your suggestion. We have revised the figure accordingly. Please see the revised one:

4. IMZ was not so readable in Figure 1.

   Response: Thanks for your comment, we have separate figure one as two to make the element such as IMZ boundary clearer.

5. The English should be further polished and improved.

   Response: Thanks for your suggestion, we have polished our MS and the certification is shown as below.

[Figure]

**Editing Certificate**

This document certifies that the manuscript

**GMIE-100: A global maximum irrigation extent and irrigation type dataset derived via irrigation performance during drought stress and machine learning methods**

prepared by the authors

**Fuyou Tian, Bingfang Wu, Hongwei Zeng, Miao Zhang, Weiwei Zhu, Nana Yan, Yuming Lu Yifan Li**

was edited for proper English language, grammar, punctuation, spelling, and overall style by one or more of the highly qualified native English speaking editors at AJE.

This certificate was issued on **March 20, 2024** and may be verified on the AJE website using the verification code **43DF-D3A8-CFB3-00BF-59DP** .

[Figure]

Neither the research content nor the authors' intentions were altered in any way during the editing process. Documents receiving this certification should be English-ready for publication; however, the author has the ability to accept or reject our suggestions and changes. To verify the final AJE edited version, please visit our verification page at aje.com/certificate. If you have any questions or concerns about this edited document, please contact AJE at support@aje.com.

AJE provides a range of editing, translation, and manuscript services for researchers and publishers around the world. For more information about our company, services, and partner discounts, please visit aje.com.

---

## Author Comment (AC2)

We are highly appreciated for your constructive comments and suggestions on our manuscript. Those comments and suggestions are valuable and helpful for revising and improving our article, as well as inspiring our research. We have carefully reviewed the comments and have revised the manuscript accordingly. Our responses are given in a point-by-point manner below and **BLUE** fonts. Please find our detailed responses in supplement to all these comments/suggestions and thank you again for everything you have contributed.

**RC3**

This manuscript introduced the GMIE-100 dataset, which identifies global irrigated cropland using drought stress performance and machine learning. This is a valuable dataset that could benefit various fields, including agriculture, environmental science, and water resource management. However, I have some major concerns about this MS that need the authors to clarify before it is further processed.

1 The title of the manuscript indicates that the dataset represents the largest irrigated area. How does the author interpret this "largest area"? This requires the author to provide explicit clarification within the text. Additionally, how does the author consider the possibility of overestimation of this largest area relative to the actual distribution, given that our focus is on the actual distribution range?

Response: Thanks for your valuable suggestion comments. The largest area should be understood separately for region with regular irrigation (RIR)and region with irregular irrigation (RIO). For RIR, the largest area means the cropland area irrigated one time at least for last three years (2017-2019). Because we detect irrigation every year for this region. To avoid missing fallow land, we identify the largest extent for last three years (2017-2019).

For RIO, it means the cropland area irrigated one time at least for last ten years (2010-2019). For RIO, irrigation occurs occasionally. We detect whether the cropland is irrigated in the driest year. But in the normal year, the irrigation maybe not necessary in this area. So, this means the largest extent area for last ten years (2010-2019).

We add this explanation in the conclusion and discussion part (Line 477-483).

As for the overestimation irrigated cropland, we can make sure that irrigation occurs one time at least in RIR for last three years (2017-2019) and in RIO for last ten years (2010-2019). In terms of principle of this method, we detect irrigation when it is necessary under water stress. On the other hand, when we compare our result with nation census data, the

result shows high consistent. Compared with USGS-LGRIP30 and GRIPC-500, our result didn't show much overestimation.

2 The samples are derived from different collection methods. It is crucial for the author to clarify whether samples collected through different methods exhibit consistent representation and describe irrigated land in the same manner. If their collection standards vary, the author needs to explicitly discuss the impact on the results.

Response: Thanks for your valuable suggestion. The representation of samples was extremely important for the final accuracy. Nevertheless, it is hard to collect the irrigation field point globally, even crop types samples. So, we fused three independent sources, the GVG field data, USGS-samples and visual interpenetration data. You can see the distribution of samples from three sources in the following figures and a specific number for each country.

[Figure]

Table 1 Number of samples in different countries and sources

| Sources | Number | Distributed country |
| --- | --- | --- |
| GVG field data | 78,338 | China(72,224) \Cambodia\Ethiopia\Zambia\ Zimbabwe |
| USGS-samples | 17,076 | Brazil (13,368), Australia (2,192), Thailand (393), and Tunisia (389) |
| VHR-interpratation | 19,965 | Rest Countries |
| total | 115,379 | |

From different country, there is varied dominant samples source. Such as in China, most of samples was obtained from GVG field survey. While in Brazil, major samples were from USGS samples. Except country with GVG and USGS-samples, the visual interpretation

data was dominant sources of samples. This also ensure the represented manner of irrigated cropland.

This could definitely introduce some uncertainty in terms of samples representatives. This effect should be acceptable in arid and semi-arid regions because the irrigation performance is relatively easy to identify. However, the uncertainty maybe enlarged in wet region due to complex manner of irrigated cropland.

We add this uncertainty of representations in the discussion part (Line 510-517) shown as below:

*"It is hard to collect the filed samples globally, we fused three sources of samples. From different country, there is varied dominant samples source. Such as in China, most of samples was obtained from GVG field survey. While in Brazil, major samples were from USGS samples. Except country with GVG and USGS-samples, the visual interpretation data was dominant sources of samples. This also ensure the represented manner of irrigated cropland. Overall, the number of samples was very large. Basically, this irrigated and rain-fed samples database could meet the globally irrigated cropland mapping compared with global cropland expansion mapping research (Potapov et al., 2022), which achieved cropland mapping globally with thousands samples.*

*Meanwhile, this fused samples maybe introduce some uncertainty in terms of representation. This effect should be acceptable in arid and semi-arid regions because the irrigation performance is relatively easy to identify. However, the uncertainty maybe enlarged in wet region due to complex manner of irrigated cropland. "*

3 In terms of accuracy assessment, merely providing overall accuracy is insufficient. Please refer to best practices for reporting accuracy as outlined in papers such as Olofsson et al. 2014 [1]. Moreover, I have not observed quantification of uncertainty, which necessitates further work from the author.

Olofsson P, Foody GM, Herold M, Stehman SV, Woodcock CE, Wulder MA. Good practices for estimating area and assessing accuracy of land change. Remote Sensing of Environment 2014; 148:42–57. https://doi.org/10.1016/j.rse.2014.02.015.

Response: Thanks for your valuable comments. We will changed all the accuracy assessment following the commended practice and evaluate the uncertainty of total area estimation.

Briefly, the overall accuracy of GIME-100 was 83.6%±0.6% with producer accuracy of 86.1%±0.7% and UA of 82.20%±0.8%. And the total area of GMIE is estimated as 403.17±9.82Mha, accounting for 23.4%±0.6% of the global cropland.

For the GCPIS data ,the overall Accuracy was 97.87%±0.1% with producer accuracy of 81.75%±0.2% and UA of 92.68%±0.1%. And the total area of GCPIS is estimated as 11.5±0.01Mha.

We have changed the statement of accuracy assessment and area estimation in the body text.

4 The results and discussion sections lack necessary citations. Many explanations proposed by the author lack corresponding literature support, which makes it difficult for me to be convinced of the correctness of your interpretations. Please see the annotations I've made in the manuscript.

Response: Thanks for your specific comments. We add necessary citation in the revised version. Please see the resubmitted version.

5 I have made several annotations in the manuscript indicating areas that need revision. It is advised that the author make corresponding modifications and carefully review the entire document to rectify similar errors.

Response: Thanks for your nice suggestion. Firstly, AJE have re-polished this MS for us, and the certification is show as below. Also, we carefully check the whole MS again and revised the similar errors. Please see the revised version.

[Figure]

Potapov, P., Turubanova, S., Hansen, M. C., Tyukavina, A., Zalles, V., Khan, A., Song, X. P., Pickens, A., Shen, Q., and Cortez, J.: Global maps of cropland extent and change show accelerated cropland expansion in the twenty-first century, Nat Food, 3, 19-28, 10.1038/s43016-021-00429-z, 2022.

---

## Author Comment (AC3)

We are highly appreciated for your constructive comments and suggestions on our manuscript. Those comments and suggestions are valuable and helpful for revising and improving our article, as well as inspiring our research. We have carefully reviewed the comments and have revised the manuscript accordingly. Our responses are given in a point-by-point manner below and **BLUE** fonts. Please find our detailed responses in supplement to all these comments/suggestions and thank you again for everything you have contributed.

**RC1**

This article describes a dataset purporting to describe maximum irrigation extent and irrigation type with global scope at a 100-metre resolution. This dataset would have broad applicability for agricultural, economic and other analyses at global and more localised levels.

The authors make an attempt at providing this dataset at such a refined resolution, however there are some fundamental issues that need to be addressed before it could actually deliver what the authors promise in the article. I believe currently the authors give a flawed sense of accuracy in their estimates of irrigated and non-irrigated land. In its current form I do not recommend this manuscript/dataset be accepted for publication in ESSD.

1. Areas and cropland definition: This dataset/manuscript needs better clarification of what areas of irrigated and non-irrigated land are included. For instance the title suggests the dataset is global, implying all irrigated and non-irrigated land are included. In the abstract they state 'In our study, we present a robust methodology that leverages irrigation performance during drought stress as an indicator of crop productivity and water consumption to identify global irrigated cropland.' The latter implies it includes only cropland. Cropland has different definitions to different authors (see Tubiello et al 2023: https://www.nature.com/articles/s43016-022-00667-9) and can be very tricky to differentiate properly. In section 2.3 the authors state they use the JECAM definition of cropland which includes land used for seasonal crops (sowed/planted and harvested at least once within the 12 months) such as cereals, root and tuber crops, oil crops as well as economically significant crops like sugar, vegetables, and cotton. Additionally land occupied by greenhouses was considered as cropland. Greenhouses in cropland is a strange inclusion and needs explaining.

> Response: Thanks for your comments. According to different definition, greenhouses belong to different class. But greenhouse is often considered part of arable land, especially in facility agriculture. They provide a controlled environment for growing a wide range of crops, thus extending the functions of traditional arable land. Greenhouses allow farmers to grow crops in areas or seasons that may not be suitable for open-air cultivation, optimizing crop growth by controlling conditions such as temperature, humidity and light. In the classification system of ChinaCover and Globalland 30, Green house was included in Cropland.

Because we used Synthesized cropland mask from ChinaCover in China, so the greenhouse was recognized as cropland in this research.

2. The authors then go on to say they used "The cropland mask at 30- meter resolution could be obtained from International Research Center of Big Data for Sustainable Development goals via https://data.casearth.cn/thematic/cbas_2022/158". They state the overall accuracy of this dataset is 89.4%, but when I look at maps from these data it appears as though they include a lot of non-cropland area esp. pasture and meadow land (see Fig 1 below). I therefore do not have confidence that this dataset is suitable for supporting the authors assertion that their dataset has 100 metre resolution.

Response: Thanks for your comments. The data was at 30 meter resolution. You could view it online via http://desp.casearth.cn/data-preview/?id=GCL30_2020&lang=en or download it via https://data.casearth.cn/en/sdo/detail/62ff50e208415d271ab1b84a.

We are sure that the accuracy of this basically acceptable. Because this data integrated 10 existing land cover maps or cropland datasets to delimit the global cropland extent while masking out irrelevant non-cropland pixels for the period of 2016–2018 (Figure 1). More detailed information on these land cover and cropland layer products as well as their classes used in the integration refer to (Zhang et al., 2021). Although variations in classification systems among different products exist, a subset of classes of those land cover and cropland layer products were selected to best fit into the cropland definition. Spatially, FROM-GLC was selected for Europe, Africa, New Zealand, the majority of Asia, and part of Latin America. GFSAD30 was selected for tropical Asian islands, including Indonesia, Malaysia, and the Philippines. In addition to these two global-coverage cropland extent products, several national or regional datasets, including ChinaCover, CDL, AAFC ACI, NLCD, MapBiomass, CLUM, SERVIR, and INTA, were used because they have been extensively validated by local experts and hence exhibited high accuracies of cropland mapping.

[Figure]

Figure 1 Spatial distribution of the land cover/cropland layer products used for the global 30-m cropland (Zhang et.al 2021 ESSD)

Zhang, M., Wu, B., Zeng, H., He, G., Liu, C., Tao, S., Zhang, Q., Nabil, M., Tian, F., Bofana, J., Beyene, A. N., Elnashar, A., Yan, N., Wang, Z., and Liu, Y.: GCI30: a global dataset of 30;m cropping intensity using multisource remote sensing imagery, Earth Syst. Sci. Data, 13, 4799-4817, 10.5194/essd-13-4799-2021, 2021.

[Figure]

Figure 1: Screenshot of casearth.cn datset.

3. Furthermore, the title of this manuscript implies this dataset is for 'maximum irrigation extent' i.e. all irrigation. They assess centre pivot irrigation, but it is not clear if the authors include lateral irrigators which is much the same technology as centre pivot, only it could be harder to distinguish lateral irrigation due to the patterns of NDVI (see figure 12).

Response: We didn't include lateral irrigation. The identification was rely on the circle shape in the satellite data. But the lateral irrigation didn't show this feature. But in the

maximum irrigation extent wen include all the irrigation types that could mitigate water stress.

4. Finally, as per section 2.1 the research relied on evapotranspiration data at a 500 m resolution. Shouldn't the authors state that the resolution of their irrigation dataset is equivalent to the lowest resolution of their input data? Otherwise you are giving a false sense of accuracy.

Response: Thanks for your comments. The evapotranspiration, precipitation product with 500 meter resolution was used to determine the driest months within each IMZ. And the time period was used to detect irrigation performance and detect irrigated cropland. In each IMZ, 30meter NDVI data was used as major input. Then to avoid effect fallow land and crop rotation, we calculate the irrigation proportion within 100meters.

5. Given the above uncertainties in cropland categorisation I suggest the authors use a definition of cropland that aligns to something like that used by the FAO. This will improve the applicability of the dataset.

Response: Thanks for your valuable comments. The crop land definition from FAO was "Cropland is land used for the cultivation of crops, both temporary (annuals) and permanent (perennials), and may include areas periodically left fallow or used as temporary pasture." Actually, we just focus temporary cropland because this was food producing crop type. The permanent crops were usually for fruit trees, nut trees, coffee, tea, and some types of vines, which is recognized as shrub or tree in most landcover system such as ESRI, FROM-GLC, GLAD-Map, GLC-FCS30 and WORDCOER. On the contrary, harvest crops, maize, soybean, wheat, and rice was most important feeding crops. So, we choose this definition to distinguish irrigated and rainfed cropland.

| CROPLAND MAP | MAPPED CATEGORY AND OPERATIONAL DEFINITIONS |
|---|---|
| ESRI[2] | **Crops.** Human planted/plotted cereals, grasses, and crops not at tree height; examples: corn, wheat, soy, fallow plots of structured land. |
| FROM-GLC[3] | **Croplands**. Land that has clear traits of intensive human activity. It varies a lot from bare field, seeding, crop growing to harvesting. It includes arable and tillage land with herbaceous/shrub crops and land with plastic foam or grass roof protection with distinguishing spectral properties. Fruit trees are classified into forests. |
| GLAD_Map[1] | **Cropland.** Land used for annual and perennial herbaceous crops for human consumption, forage (including hay) and biofuel. Perennial woody crops, permanent pastures and shifting cultivation are excluded. |
| GLC-FCS30[4] | **Cropland**. Rainfed and irrigated cropland. Detailed (Level 2) data on herbaceous cover. In this analysis, we excluded detailed data on Tree or shrub cover for better comparison with other layers. |
| GLOBELAND30[5] | **Cropland**. Category includes paddy fields, irrigated dry land, rain-fed dry land, vegetable land, pasture planting land, greenhouse land, land mainly for planting crops with fruit trees and other economic trees, as well as tea gardens, coffee gardens and other shrubs. |
| WORLDCOVER[6] | **Cropland**. Land covered with annual cropland that is sowed/planted and harvestable at least once within the 12 months after the sowing/planting date. The annual cropland produces an herbaceous cover and is sometimes combined with some tree or woody vegetation. Note that perennial woody crops will be classified as the appropriate tree cover or shrub land cover type. Greenhouses are considered as built-up. |

Figure 2 Definition of Cropland in mainstream landcover according to Tubiello et.al 2023

Tubiello F N, Conchedda G, Casse L, et al. Measuring the world's cropland area[J]. Nature Food, 2023, 4(1): 30-32.

6. This manuscript needs to be edited heavily before it is resubmitted. I made a note of some of these edits in minor comments in the first few pages. Note, the list I provide is not exhaustive as there were many other changes to make.

   Response: Thanks for your valuable comments. We have carefully checked and polished our MS and the certification is shown as below:

[Figure]

**Editing Certificate**

This document certifies that the manuscript

**GMIE-100: A global maximum irrigation extent and irrigation type dataset derived via irrigation performance during drought stress and machine learning methods**

prepared by the authors

**Fuyou Tian, Bingfang Wu, Hongwei Zeng, Miao Zhang, Weiwei Zhu, Nana Yan, Yuming Lu Yifan Li**

was edited for proper English language, grammar, punctuation, spelling, and overall style by one or more of the highly qualified native English speaking editors at AJE.

This certificate was issued on **March 20, 2024** and may be verified on the AJE website using the verification code **43DF-D3A8-CFB3-00BF-59DP** .

[Figure]

Neither the research content nor the authors' intentions were altered in any way during the editing process. Documents receiving this certification should be English-ready for publication; however, the author has the ability to accept or reject our suggestions and changes. To verify the final AJE edited version, please visit our verification page at aje.com/certificate. If you have any questions or concerns about this edited document, please contact AJE at support@aje.com.

AJE provides a range of editing, translation, and manuscript services for researchers and publishers around the world. For more information about our company, services, and partner discounts, please visit aje.com.

7. Lines 134-140. A better plain language description of how irrigated and non-irrigated land was categorised is needed.

Response: Thanks for your valuable comments. Inspired from purpose of irrigation, what is to mitigate the effect of water stress. Basically, we assume that water stress can be regular or irregular. If there is crops during dry season, the irrigation should occurs regular. Otherwise, irrigation is just complementary to rainfall in extremely dry year, which means irrigation is irregular. For regular irrigation, we could detect vegetation signal in the dry season (DM-NDVI) when precipitation couldn't meet water demand for crops. For irregular irrigation, we compare the NDVI in extremely dry year with 10-year average level and calculate the deviation($NDVI_{dev}$) to determine whether it is irrigated or not. To determine whether, it is region with regular or irregular irrigation, we used both of these two indicators and choose the method get higher accuracy.

We have change the explanation in the body text.

8. Section 3.4. The uncertainty in estimates of cropland used in the authors models needs to be better explained. Differences in classification of 'cropland' for instance can contribute to variation in estimates in irrigated cropland mentioned in section 3.1.

Response: Thanks for your valuable comments. We have include this uncertainty in the discussion part.
"The crop land definition from FAO was "Cropland is land used for the cultivation of

crops, both temporary (annuals) and permanent (perennials), and may include areas periodically left fallow or used as temporary pasture." Actually, we just focus temporary cropland because this was food producing crop type. The permanent crops were usually for fruit trees, nut trees, coffee, tea, and some types of vines, which is recognized as shrub or tree in most landcover system such as ChinaCover. On the contrary, harvest crops, maize, soybean, wheat, and rice was most important feeding crops. These may lead to uncertainty for some extent"

  Also, we add uncertainty assessment of irrigated cropland area estimation. We change the absolute value to confidence level of 403.17±9.82 Mha

Minor comments

1. Abstract 1st line 11. "primary sector of human water…"; Use other word than sector such as form.

   Response: Thanks for your valuable comments, we changed the sentence to *"Irrigation accounts for the major form of human water consumption and plays a pivotal role in enhancing crop yields and mitigating drought effects."*.

2. Line 26: What is the DL method? Define when you first use an abbreviation.

   Response: Thanks for your detail comments, DL means deep learning method, we change this sentence to "Furthermore, with the help of the deep learning (DL) method, the global central pivot irrigation system (CPIS) was identified using Pivot-Net"

3. Line 27: What is Pivot-Net?

   Response: Thanks for your detail comments, It means a novel convolutional neural network based on U-net. We added it in the text.

4. Line 29: "The GMIE-100 dataset containing both or irrigated extent…". What does the both relate to?

   Response: Thanks for your detail comments. It should be "The GMIE-100 dataset containing both the irrigated extent and CPIS distribution"

5. Line 40 use reference to back up claim that highest resolution maps are 500m to 10km.

   Response: Thanks for your detail comments, we have add three reference for this sentence.

6. Line 60 use space between croplands and (Thenkabail et al 2009)

Response: Thanks for your detail comments. We have change it accordingly.

7. Line 106. Use reference to back up claim of 80% efficiency.

   Response: Thanks for your detail comments, We have added the citation.

8. Throughout references and tables, make sure abbreviations are defined esp in title of Figure 1 and 2.

   Response: Thanks for your detail comments. We have added all the abbreviation in the tile.

9. Line 175. What is GVG?

   Response: Thanks for your detail comments. GVG (GPS, Video, GIS) application serves as a comprehensive field data collection system that integrates GPS for precise positioning, a video for capturing geo-tagged photographs, and a GIS system for managing geographic information. You could download it via https://gvgserver.cropwatch.com.cn/download . We added this information in the body text.

10. Line 251. Spelling mistakes in Nirrgated and Nnon-irrgated.

    Response: Thanks for your detail comments, we have corrected it.

11. Line 268. Spelling mistake exemple.

    Response: Thanks for your detail comments, we have corrected it.

12. Line 377: belt_Mexican coastal plain. Error.

    Response: Thanks for your detail comments, we have corrected it.

13. Line 469-471: How does looking at if an area of land has been cultivated during the driest month over a span of three year help determine if it is irrigated land? What if the cultivation occurs in one of the regular wet seasons of the year but irrigated is still needed thereafter?

    Response: Thanks for your comments. In the first case it should be irrigated. As for the second case, if the cultivation occurs in one of the regular wet seasons of the year but irrigated is still needed thereafter, we need to see whether water could meet crop requirement in another growing season. If there is a regular water stress in thereafter growing season, it is region with regular irrigation. Otherwise, it is region needs irrigation occasionally.

---

## Author Response (AR1)

We are highly appreciated for your constructive comments and suggestions on our manuscript. Those comments and suggestions are valuable and helpful for revising and improving our article, as well as inspiring our research. We have carefully reviewed the comments and have revised the manuscript accordingly. Our responses are given in a point-by-point manner below and **BLUE** fonts. Please find our detailed responses in supplement to all these comments/suggestions and thank you again for everything you have contributed.

**RC1**

This article describes a dataset purporting to describe maximum irrigation extent and irrigation type with global scope at a 100-metre resolution. This dataset would have broad applicability for agricultural, economic and other analyses at global and more localised levels.

The authors make an attempt at providing this dataset at such a refined resolution, however there are some fundamental issues that need to be addressed before it could actually deliver what the authors promise in the article. I believe currently the authors give a flawed sense of accuracy in their estimates of irrigated and non-irrigated land. In its current form I do not recommend this manuscript/dataset be accepted for publication in ESSD.

Response: Thanks for your overall comments. As we all know, irrigation is important for food producing and water resource management. But the updated and high-resolution irrigation dataset is still rare. We believe this dataset will promote the understanding of global irrigation distribution and support the related application.

The most concerned points, as you mentioned, was the cropland extent. Actually, we already used almost the state-of-art landcover for each region to synthesized a global cropland mask. **These data have been utilized for their extensive validation by local experts, usually leading to their high precision in mapping cropland.** The cropland mask used in this research integrated more than 10 cropland dataset including global cropland product: FROM-GLC, GFSAD30 as well as National and regional data sets, such as ChinaCover (Wu et al., 2017; Wu et al., 2024), Cropland Data Layers (Boryan et al., 2011), Agriculture and Agri-Food Canada Annual Crop Inventory (Fisette et al., 2013; Mcnairn et al., 2009), MapBiomass (Do Canto et al., 2020) et.al. Please see the detail reply below.

**Q1.** Areas and cropland definition: This dataset/manuscript needs better clarification of what areas of irrigated and non-irrigated land are included. For instance the title suggests the dataset is global, implying all irrigated and non-irrigated land are included. In the abstract they state 'In our study, we present a robust methodology that leverages irrigation performance during drought stress as an indicator of crop productivity and water consumption to identify global irrigated cropland.' The latter implies it includes only cropland. Cropland has different definitions to different authors (see Tubiello et al 2023: https://www.nature.com/articles/s43016-022-00667-9) and can be very tricky to

differentiate properly. In section 2.3 the authors state they use the JECAM definition of cropland which includes land used for seasonal crops (sowed/planted and harvested at least once within the 12 months) such as cereals, root and tuber crops, oil crops as well as economically significant crops like sugar, vegetables, and cotton. Additionally land occupied by greenhouses was considered as cropland. Greenhouses in cropland is a strange inclusion and needs explaining. The authors then go on to say they used "The cropland mask at 30- meter resolution could be obtained from International Research Center of Big Data for Sustainable Development goals via https://data.casearth.cn/thematic/cbas_2022/158". They state the overall accuracy of this dataset is 89.4%, but when I look at maps from these data it appears as though they include a lot of non-cropland area esp. pasture and meadow land (see Fig 1 below). I therefore do not have confidence that this dataset is suitable for supporting the authors assertion that their dataset has 100 metre resolution.

Response: Thanks for your comments.

We agree with you that the cropland mask have crucial effect for the final result. Due to lack of high-resolution and consistent cropland data layer, we used synthesized data layer to depict the cropland extent. This data integrated more than 10 cropland dataset including global cropland product: FROM-GLC, GFSAD30 as well as National and regional data sets, such as ChinaCover (Wu et al., 2024; Wu et al., 2017), Cropland Data Layers (Boryan, Yang, Mueller, & Craig, 2011), Agriculture and Agri-Food Canada Annual Crop Inventory (Fisette et al., 2013; McNairn, Champagne, Shang, Holmstrom, & Reichert, 2009), MapBiomass (do Canto et al., 2020) et.al for the period of 2016–2018. **These data have been utilized for their extensive validation by local experts, usually leading to a high precision in mapping cropland (Wu et al., 2023).**

The detail information for the source of cropland mask was listed in Table S1 and Figure S1. Spatially, FROM-GLC was selected for Europe, Africa, New Zealand, the majority of Asia, and part of Latin America. GFSAD30 was selected for tropical Asian islands, including Indonesia, Malaysia, and the Philippines (Figure S1). In addition to these two global-coverage cropland extent products, several national or regional datasets, including ChinaCover, CDL, AAFC ACI, NLCD, MapBiomass, CLUM, SERVIR, and INTA.

Although variations in classification systems among different products exist, a subset of classes of those land cover and cropland layer products were selected to best fit into the cropland definition (Table S1).

The data was at 30 meter resolution, which can be viewed online via http://desp.casearth.cn/data-preview/?id=GCL30_2020&lang=en or downloaded via https://data.casearth.cn/en/sdo/detail/62ff50e208415d271ab1b84a.This data was present to United Nations on behalf of the Chinese government by Wang Yi, Foreign Minister of China (https://www.mfa.gov.cn/eng/zxxx_662805/202209/t20220922_10769737.html). We are sure that the accuracy of this synthesized cropland mask was basically acceptable. This data was used for supporting crop intensity mapping (Zhang et al., 2021).

According to different classification system, greenhouses belong to different class. But greenhouse is often considered part of arable land, especially in facility agriculture. Greenhouses allow farmers to grow crops in areas or seasons that may not be suitable for open-air cultivation, optimizing crop growth by controlling conditions such as temperature, humidity and light. In the classification system of ChinaCover (Wu et al., 2024; Wu et al., 2017) and Globalland 30 (Chen et al., 2015), Green house was included in Cropland. Because we used Synthesized cropland mask from ChinaCover in China, so the greenhouse was recognized as cropland in this research.

The following text has been added:

Line 258-234:

*This data integrated more than 10 cropland dataset including global cropland product: FROM-GLC, GFSAD30 as well as National and regional data sets, such as ChinaCover (Wu et al., 2017; Wu et al., 2024), Cropland Data Layers (Boryan et al., 2011), Agriculture and Agri-Food Canada Annual Crop Inventory (Fisette et al., 2013; Mcnairn et al., 2009), MapBiomass (Do Canto et al., 2020) et.al. More information about this cropland mask can be found in supplementary. These data have been utilized for their extensive validation by local experts, leading to their high precision in mapping cropland (Wu et al., 2023a)*

Line 550-557:

*Actually, we just focus on seasonal cropland, because the permanent crops were usually for fruit trees, nut trees, coffee, tea, and some types of vines, which is recognized as shrub or tree in most landcover system such as ESRI (Karra et al., 2021), FROM-GLC (Yu et al., 2013), GLAD_Map (Potapov et al., 2022), GLC-FCS30 (Zhang et al., 2021b) and WORDCOER (Zanaga et al., 2022). On the contrary, harvest crops, maize, soybean, wheat, and rice was most important for food security. So, we choose this definition to distinguish irrigated and rainfed cropland, rather than the definition from FAO's. Different definition of crop as input data may produce varied irrigated cropland area, which will definitely introduce uncertainty in the final result. A consistent, high resolution cropland mask with high accuracy is urgently needed to solve this problem.*

We provided this explanation in the supplementary materials to clarify the reliability of this cropland Mask.

[Figure]

Figure S1 Spatial distribution of the land cover/cropland layer products used for the global 30-m cropland (Zhang et.al 2021 ESSD)

Table S1. Cropland and land cover datasets used for the study

| Region | Dataset name | Year | Selected classes | Resolution | Accuracy | Reference |
|---|---|---|---|---|---|---|
| Argentina | Crop type map | 2018-19 | | 30 | 81% | Abelleyra et al, 2019(de Abelleyra Diego, 2019) |
| Australia | Catchment Scale Land Use of Australia | 2018 | Cropping, Seasonal horticulture, Irrigated cropping, and Irrigated seasonal horticulture | 50 | 0.92% | ABARES, 2016 |
| Brazil | MapBiomas | | | 30 | | Project MapBiomas, 2019* |
| Bhutan | Land cover data of Bhutan | 2010 | Agriculture | 30 | | ICIMOD, 2011 |
| Canada | Canada AAFC Annual Crop Inventory data | 2009 | Seasonal crops and greenhouse | 30 | 85% | (McNairn, H,2009) (McNairn et al., 2009) |
| China | ChinaCover | 2015 | Upland and rice field | 30 | 86% | Wu et al., 2017 |
| Mozambique | ChinaCover | 2018 | | 10 | 85% | Bofana et al., 2020 |
| Nepal | National landcover for Nepal | 2010 | Agriculture area | 30 | | Uddin et al., 2015(Uddin et al., 2015) |
| New Zealand | New Zealand Land Cover | 2012 | Short-rotation cropland | 30 | | NZLRI, 2015 |
| United States | CDL | 2009 | Class 1~56 and Class 225~254 | 30 | 85%-95% | Boryan, C.,et al. 2011 |
| Zambia | ChinaCover | 2018 | | 10 | 0.87 | Bofana et al., 2020 |
| Zimbabwe | ChinaCover | 2018 | | 10 | 0.86 | Bofana et al., 2020 |
| Europe | CORINE land cover | 2018 | | | | Büttner, et al., 2017(Büttner, Kosztra, Soukup, Sousa, & Langanke, 2017) |
| Central Asia | CA Landcover | 2015 | | 30 | | CASEarth |
| Africa | FROM-GLC-Africa30 | 2015 | | | | Feng et al., 2018 |
| Lower Mekong | SERVIR-Mekong Land Cover | 2018 | Cropland and Rice | 30 | 0.94 | Saah et al., 2020(Saah et al., 2020) |
| Global | FORM-GLC 2015 | 2015 | | 30 | | ** |
| Global | GFSAD30 | 2015 | | 30 | | *** |
| * | https://plataforma.mapbiomas.org/map#coverage | | | | | |

| ** | http://www.chinageoss.org/tansat/pdf/FROM-GLC.pdf |
|---|---|
| *** | https://lpdaac.usgs.gov/news/release-of-gfsad-30-meter-cropland-extent-products/ |

Boryan, C., Yang, Z., Mueller, R., & Craig, M. (2011). Monitoring US agriculture: the US department of agriculture, national agricultural statistics service, cropland data layer program. *Geocarto International, 26*(5), 341-358.

Büttner, G., Kosztra, B., Soukup, T., Sousa, A., & Langanke, T. (2017). CLC2018 technical guidelines. *European Environment Agency, Wien*.

Chen, J., Chen, J., Liao, A., Cao, X., Chen, L., Chen, X., . . . Lu, M. (2015). Global land cover mapping at 30 m resolution: A POK-based operational approach. *ISPRS Journal of Photogrammetry and Remote Sensing, 103*, 7-27.

de Abelleyra Diego, B. S., Verón Santiago, Mosciaro Jesús, Volante José,. (2019). Mapa Nacional de Cultivos campaña 2018/2019. Retrieved from https://inta.gob.ar/sites/default/files/mapa_nacional_de_cultivos_campana_2018_2019.pdf

do Canto, A. C. B., Marques, R., Leite, F. F. G. D., da SILVEIRA, J., DONAGEMMA, G., & RODRIGUES, R. (2020). *Land use and cover maps for Mato Grosso from 1985 to 2019*.

Fisette, T., Rollin, P., Aly, Z., Campbell, L., Daneshfar, B., Filyer, P., . . . Jarvis, I. (2013). *AAFC annual crop inventory.* Paper presented at the 2013 Second International Conference on Agro-Geoinformatics (Agro-Geoinformatics).

Gorelick, N., Hancher, M., Dixon, M., Ilyushchenko, S., Thau, D., & Moore, R. (2017). Google Earth Engine: Planetary-Scale Geospatial Analysis for Everyone. *Remote Sensing of Environment, 202*, 18-27.

McNairn, H., Champagne, C., Shang, J., Holmstrom, D., & Reichert, G. (2009). Integration of optical and Synthetic Aperture Radar (SAR) imagery for delivering operational annual crop inventories. *ISPRS Journal of Photogrammetry and Remote Sensing, 64*(5), 434-449. doi:10.1016/j.isprsjprs.2008.07.006

Saah, D., Tenneson, K., Poortinga, A., Nguyen, Q., Chishtie, F., San Aung, K., . . . Cutter, P. (2020). Primitives as building blocks for constructing land cover maps. *International Journal of Applied Earth Observation and Geoinformation, 85*, 101979.

Tubiello, F. N., Conchedda, G., Casse, L., Pengyu, H., Zhongxin, C., De Santis, G., . . . Muchoney, D. (2023). Measuring the world's cropland area. *Nature Food, 4*(1), 30-32.

Uddin, K., Shrestha, H. L., Murthy, M., Bajracharya, B., Shrestha, B., Gilani, H., . . . Dangol, B. (2015). Development of 2010 national land cover database for the Nepal. *Journal of Environmental Management, 148*, 82-90.

Wu, B., Fu, Z., Fu, B., Yan, C., Zeng, H., & Zhao, W. (2024). Dynamics of land cover changes and driving forces in China's drylands since the 1970 s. *Land Use Policy,*

*140*, 107097. doi:10.1016/j.landusepol.2024.107097

Wu, B., Qian, J., Zeng, Y., Zhang, L., Yan, C., Wang, Z., . . . Huang, J. (2017). Land Cover Atlas of the People's Republic of China (1: 1 000 000). *Science Bulletin, 65*, 1125-1136.

Wu, B., Tian, F., Nabil, M., Bofana, J., Lu, Y., Elnashar, A., . . . Zhu, W. (2023). Mapping global maximum irrigation extent at 30m resolution using the irrigation performances under drought stress. *Global Environmental Change, 79*, 102652. doi:10.1016/j.gloenvcha.2023.102652

Zhang, M., Wu, B., Zeng, H., He, G., Liu, C., Tao, S., . . . Liu, Y. (2021). GCI30: a global dataset of 30 m cropping intensity using multisource remote sensing imagery. *Earth Syst. Sci. Data, 13*(10), 4799-4817. doi:10.5194/essd-13-4799-2021

**Q2.** Furthermore, the title of this manuscript implies this dataset is for 'maximum irrigation extent' i.e. all irrigation. They assess centre pivot irrigation, but it is not clear if the authors include lateral irrigators which is much the same technology as centre pivot, only it could be harder to distinguish lateral irrigation due to the patterns of NDVI (see figure 12).

Response: The identification was relied on the circle shape in the satellite data. So, we didn't include lateral irrigation. But the lateral irrigation didn't show this feature. But in the maximum irrigation extent we include all the irrigation types that could mitigate water stress.

The following text has been added:

Line 522-524:
*However, this study didn't include the lateral irrigation, because the identification of irrigation method was relied on the circle shape in the satellite data and the lateral irrigation didn't show this feature. In the maximum irrigation extent, we include all the irrigation types that could mitigate water stress.*

**Q3.** Finally, as per section 2.1 the research relied on evapotranspiration data at a 500 m resolution. Shouldn't the authors state that the resolution of their irrigation dataset is equivalent to the lowest resolution of their input data? Otherwise you are giving a false sense of accuracy.

Response: Thanks for your comments. The evapotranspiration, precipitation product with 500-meter resolution was used to determine the driest months within each IMZ. And the time period was used to detect irrigation performance and detect irrigated cropland. In each IMZ, 30meter NDVI data was used as major input. Then to avoid effect fallow land and crop rotation, we calculate the irrigation proportion within 100 meters.

The following text has been added:

Line 500-503:
*The evapotranspiration, precipitation product with 500-meter resolution was used to determine the driest months within each IMZ. And the time period was used to detect irrigation performance and detect irrigated cropland. In each IMZ, 30-meter NDVI data was used as major input. Then to avoid effect fallow land and crop rotation, we calculate the irrigation proportion within 100 meters.*

**Q4.** Given the above uncertainties in cropland categorisation I suggest the authors use a definition of cropland that aligns to something like that used by the FAO. This will improve the applicability of the dataset.

Response: Thanks for your valuable comments.

The crop land definition from FAO was "Cropland is land used for the cultivation of crops, both temporary (annuals) and permanent (perennials), and may include areas periodically

left fallow or used as temporary pasture." Actually, we just focus temporary cropland because this was food producing crop type. The permanent crops were usually for fruit trees, nut trees, coffee, tea, and some types of vines, which is recognized as shrub or tree in most landcover system such as ESRI, FROM-GLC, GLAD-Map, GLC-FCS30 and WORDCOER. On the contrary, harvest crops, maize, soybean, wheat, and rice was most important feeding crops.

So, we choose this definition to distinguish irrigated and rainfed cropland. As mentioned by Francesco et.al in *Measuring Measuring the world's cropland area* (Tubiello et al., 2023), the cropland mask in most remote sensing products were more closer to the definition of **arable cropland** from FAO. He also recommended to use the correct FAO terminology to avoid confusion. The permanent crops are a FAO sub-category that is likely to be classified as grassland, rather than cropland, in most remote sensing products.

| CROPLAND MAP | MAPPED CATEGORY AND OPERATIONAL DEFINITIONS |
|---|---|
| ESRI[2] | **Crops.** Human planted/plotted cereals, grasses, and crops not at tree height; examples: corn, wheat, soy, fallow plots of structured land. |
| FROM-GLC[3] | **Croplands.** Land that has clear traits of intensive human activity. It varies a lot from bare field, seeding, crop growing to harvesting. It includes arable and tillage land with herbaceous/shrub crops and land with plastic foam or grass roof protection with distinguishing spectral properties. Fruit trees are classified into forests. |
| GLAD_Map[1] | **Cropland.** Land used for annual and perennial herbaceous crops for human consumption, forage (including hay) and biofuel. Perennial woody crops, permanent pastures and shifting cultivation are excluded. |
| GLC-FCS30[4] | **Cropland.** Rainfed and irrigated cropland. Detailed (Level 2) data on herbaceous cover. In this analysis, we excluded detailed data on Tree or shrub cover for better comparison with other layers. |
| GLOBELAND30[5] | **Cropland.** Category includes paddy fields, irrigated dry land, rain-fed dry land, vegetable land, pasture planting land, greenhouse land, land mainly for planting crops with fruit trees and other economic trees, as well as tea gardens, coffee gardens and other shrubs. |
| WORLDCOVER[6] | **Cropland.** Land covered with annual cropland that is sowed/planted and harvestable at least once within the 12 months after the sowing/planting date. The annual cropland produces an herbaceous cover and is sometimes combined with some tree or woody vegetation. Note that perennial woody crops will be classified as the appropriate tree cover or shrub land cover type. Greenhouses are considered as built-up. |

Figure 2 Definition of Cropland in mainstream landcover according to Tubiello et.al 2023 Tubiello F N, Conchedda G, Casse L, et al. Measuring the world's cropland area[J]. Nature Food, 2023, 4(1): 30-32.

The following text has been added:

Line 550-557:

*Actually, we just focus on seasonal cropland, because the permanent crops were usually for fruit trees, nut trees, coffee, tea, and some types of vines, which is recognized as shrub*

*or tree in most landcover system such as ESRI (Karra et al., 2021), FROM-GLC (Yu et al., 2013), GLAD_Map (Potapov et al., 2022), GLC-FCS30 (Zhang et al., 2021b) and WORDCOER (Zanaga et al., 2022). On the contrary, harvest crops, maize, soybean, wheat, and rice was most important for food security.   So, we choose this definition to distinguish irrigated and rainfed cropland, rather than the definition from FAO's. Different definition of crop as input data may produce varied irrigated cropland area, which will definitely introduce uncertainty in the final result. A consistent, high resolution cropland mask with high accuracy is urgently needed to solve this problem.*

**Q5.** This manuscript needs to be edited heavily before it is resubmitted. I made a note of some of these edits in minor comments in the first few pages. Note, the list I provide is not exhaustive as there were many other changes to make.

Response: Thanks for your valuable comments.

We have carefully checked and polished our MS. The polish certification from AJE is shown as below:

[Figure]

**Q6.** Lines 134-140. A better plain language description of how irrigated and non-irrigated land was categorised is needed.

Response: Thanks for your valuable comments.

Inspired from purpose of irrigation, what is to mitigate the effect of water stress. Basically, we assume that water stress can be regular or irregular. If there are crops during dry season, the irrigation should occur regular. Otherwise, irrigation is just complementary to rainfall

in extremely dry year, which means irrigation is irregular. For regular irrigation, we could detect vegetation signal in the dry season (DM-NDVI) when precipitation couldn't meet water demand for crops. For irregular irrigation, we compare the NDVI in extremely dry year with 10-year average level and calculate the deviation ($NDVI_{dev}$) to determine whether it is irrigated or not. To determine whether, it is region with regular or irregular irrigation, we used both of these two indicators and choose the method get higher accuracy.

The following text has been added:

Lin 134-140:

*Inspired from purpose of irrigation, what is to mitigate the effect of water stress. Basically, we assume that water stress can be regular or irregular. If there are crops during dry season, the irrigation should occur regular. Otherwise, irrigation is just complementary to rainfall in extremely dry year, which means irrigation is irregular. For regular irrigation, we could detect vegetation signal in the dry season (DM-NDVI) when precipitation couldn't meet water demand for crops. For irregular irrigation, we compare the NDVI in extremely dry year with 10-year average level and calculate the deviation ($NDVI_{dev}$) to determine whether it is irrigated or not. To determine whether, it is region with regular or irregular irrigation, we used both of these two indicators and choose the method get higher accuracy.*

**Q7.** Section 3.4. The uncertainty in estimates of cropland used in the authors models needs to be better explained. Differences in classification of 'cropland' for instance can contribute to variation in estimates in irrigated cropland mentioned in section 3.1.

Response: Thanks for your valuable comments.

The cropland masks had the greatest influence on the GMIE-100 dataset. Different definition of crop as input data may produce varied irrigated cropland area, which will definitely introduce uncertainty in the final result. A consistent, high resolution cropland mask with high accuracy is urgently needed to solve this problem."

The following text has been added:

Line 542-557:

*"The cropland masks had the greatest influence on the GMIE-100 dataset (Salmon et al., 2015; Meier et al., 2018), despite the selection of 16 distinct cropland datasets derived from country- and region-level sources as high-priority inputs. These datasets often exhibit disparities in estimating the distribution of cropland, particularly in African countries, due to the complex landscape, frequent cloud cover, and the presence of small agricultural fields (Nabil et al., 2020). Consequently, inaccuracies within the cropland datasets were transposed onto the GMIE-100 dataset. Nevertheless, importantly, these datasets remain the primary sources of cost-effective and up-to-date information covering vast geographical areas. Actually, we just focus on seasonal cropland, because the permanent*

*crops were usually for fruit trees, nut trees, coffee, tea, and some types of vines, which is recognized as shrub or tree in most landcover system such as ESRI (Karra et al., 2021), FROM-GLC (Yu et al., 2013), GLAD_Map (Potapov et al., 2022), GLC-FCS30 (Zhang et al., 2021b) and WORDCOER (Zanaga et al., 2022). On the contrary, harvest crops, maize, soybean, wheat, and rice was most important for food security. So, we choose this definition to distinguish irrigated and rainfed cropland, rather than the definition from FAO's. Different definition of crop as input data may produce varied irrigated cropland area, which will definitely introduce uncertainty in the final result. A consistent, high resolution cropland mask with high accuracy is urgently needed to solve this problem.*"

We also evaluate the uncertainty of total area estimation. The total area of GMIE is estimated as 403.17±9.82Mha, accounting for 23.4%±0.6% of the global cropland. We change all the statement across the whole text.

**Minor comments**

**Q1.** Abstract 1st line 11. "primary sector of human water…"; Use other word than sector such as form.

Response: Thanks for your valuable comments.

we changed the sentence in line 11-12:

*"Irrigation accounts for the major form of human water consumption and plays a pivotal role in enhancing crop yields and mitigating drought effects."*.

**Q2.** Line 26: What is the DL method? Define when you first use an abbreviation.

Response: Thanks for your detail comments.

DL means deep learning method. We change this sentence in Lin 25-26:

"*Furthermore, with the help of the deep learning (DL) method, the global central pivot irrigation system (CPIS) was identified using Pivot-Net, a novel convolutional neural network based on U-net.*"

**Q3.** Line 27: What is Pivot-Net?

Response: Thanks for your detail comments.

It means a novel convolutional neural network based on U-net.

We added it in in Lin 25-27:

"*Furthermore, with the help of the deep learning (DL) method, the global central pivot irrigation system (CPIS) was identified using Pivot-Net, a novel convolutional neural network based on U-net.*"

**Q4.** Line 29: "The GMIE-100 dataset containing both or irrigated extent…". What does the both relate to?

Response: Thanks for your detail comments.

We have corrected it in Line 30:

"*The GMIE-100 dataset containing both the irrigated extent and CPIS distribution*"

**Q5.** Line 40 use reference to back up claim that highest resolution maps are 500m to 10km.

Response: Thanks for your detail comments, we have added three references for this sentence.

Line 38-40:

*However, the highest available resolution for existing irrigation maps remains within a range of 500 metres to 10 kilometres (Nagaraj, Proust, Todeschini, Rulli, & D'Odorico, 2021; Siebert et al., 2005; Siebert, Henrich, Frenken, & Burke, 2013).*

**Q6.** Line 60 use space between croplands and (Thenkabail et al 2009)

Response: Thanks for your detail comments. We have change it accordingly and check all this kind of error through the whole MS.

**Q7.** Line 106. Use reference to back up claim of 80% efficiency.

Response: Thanks for your detail comments.

We have added the citation in Line 107:

*Furthermore, considerable variations in irrigation efficiency are apparent among different irrigation types, with central pivot irrigation systems (CPISs), which have an efficiency rate exceeding 80%, emerging as the predominant global sprinkler irrigation method (Tian et al., 2023)*

**Q8.** Throughout references and tables, make sure abbreviations are defined in title of Figure 1 and 2.

Response: Thanks for your detail comments.

We have added all the abbreviation in the tile.

Line 130:

*Figure 1 Samples of irrigated, rainfed and central pivot irrigation system (CPIS) from multiple sources and mapping units for irrigation mapping and CPIS identification. GVG means GPS, Video, GIS system for collecting field data. VHR means very high resolution. IMZs means Irrigation mapping zones.*

Line 146:

*Figure 2 Flow chart of GMIE-100 with a typical irrigation type of CPIS. GVG means GPS, Video, GIS system for collecting field data. VHR means very high resolution. IMZs means Irrigation mapping zones. NDVIdev : NDVI deviation in extremely dry year with 10-year average level. DM-NDVI: NDVI in the dry season.*

Line 430:

*Figure 12 Accuracy for countries with GVG (GPS, Video, GIS) irrigation validation points*

**Q9.** Line 175. What is GVG?

Response: Thanks for your detail comments. GVG (GPS, Video, GIS) application serves as a comprehensive field data collection system that integrates GPS for precise positioning, a video for capturing geo-tagged photographs, and a GIS system for managing geographic information. You could download it via https://gvgserver.cropwatch.com.cn/download.

We added more explain in Line 181-184:

*GVG (GPS, Video, GIS) application serves as a comprehensive field data collection system that integrates GPS for precise positioning, a video for capturing geo-tagged photographs, and a GIS system for managing geographic information. You could download it via https://gvgserver.cropwatch.com.cn/download.*

**Q10.** Line 251. Spelling mistakes in Nirrgated and Nnon-irrgated.

Response: Thanks for your detail comments, we have corrected it.

**Q11.** Line 268. Spelling mistake exemple.

Response: Thanks for your detail comments, we have corrected it.

**Q12.** Line 377: belt_Mexican coastal plain. Error.

Response: Thanks for your detail comments, we have corrected it.

**Q13.** Line 469-471: How does looking at if an area of land has been cultivated during the driest month over a span of three-year help determine if it is irrigated land? What if the cultivation occurs in one of the regular wet seasons of the year but irrigated is still needed thereafter?

Response: Thanks for your comments. In the first case it should be irrigated. As for the second case, if the cultivation occurs in one of the regular wet seasons of the year but irrigated is still needed thereafter, we need to see whether water could meet crop requirement in another growing season. If there is a regular water stress in thereafter growing season, it is region with regular irrigation. Otherwise, it is region needs irrigation occasionally.

**RC2**

This study demonstrated a global irrigation dataset with 100 meters using irrigation performance during drought stress, which is a brand-new way to detect irrigated and non-irrigated cropland. Furthermore, this MS finishes mapping the global central pivot irrigation system using the Deep Learning method. Also, it is interesting to detect special irrigation methods using deep learning methods. Overall, the MS was well-written and designed for readers. But there were still some concerns before this MS was accepted:

Response: Thanks for your positive comments.

Major concerns:

**Q1.** About the resolution: In section 2.1 some coarse data was described as input data, but the final resolution of the irrigation map is 100 meters, so this will mislead some readers on how to produce a 100-meter irrigation map using 500-meter data.

Response: Thanks for your comments.

The evapotranspiration, precipitation product with 500-meter resolution was used to determine the driest months within each IMZ. And the time period was used to detect irrigation performance and detect irrigated cropland. In each IMZ, 30-meter NDVI data was used as major input. Then to avoid effect fallow land and crop rotation, we calculate the irrigation proportion within 100 meters.

We also added this statement in the body text.

Line 500-503:

*The evapotranspiration, precipitation product with 500-meter resolution was used to determine the driest months within each IMZ. And the time period was used to detect irrigation performance and detect irrigated cropland. In each IMZ, 30-meter NDVI data was used as major input. Then to avoid effect fallow land and crop rotation, we calculate the irrigation proportion within 100 meters.*

**Q2.** About the IMZs: You mention that "65 MRUS in Cropwatch served as the basis for further division of global cropland into 110 irrigation mapping zones (IMZs)", what is the principle for further dividing these zones? Are these zones available or not?

Response: Thanks for your comments.

We further divided 65 zones into 110 based on arid indices, water availability, soil types, and landforms. This data is publicly available on the CropWatch website or you can contact us via email. We added it in available data source.

*The irrigation unit zone can be downloaded from http://cloud.cropwatch.com.cn/*

**Q3.** About accuracy assessment: you collect many field points using the GVG app. How to distinguish irrigation field points during the field survey?

Response: Thanks for your constructive suggestions.

Although it is not easy to identify irrigated cropland on satellite data, irrigation cropland could be identified accurate in field according to irrigation infrastructure, crop type and crop health condition. Even you cannot distinguish them following above characteristics, you could ask local farmer, who will answer this question with hesitate.

● Irrigation infrastructure, some obvious feature was easy to identify, such as canner, irrigation plump and central pivot irrigation system. We display serval photos for this case as below:

[Figure]

| Irrigation cannel in Xinjiang | Drip irrigation in Hebei province |
| Irrigation pump | Central pivot irrigation system |

- Usually, irrigated was applied for certain crop types, such as winter wheat in North China Plain, Cotton in Xinjiang and vegetable and tomatoes in most province, et.al.
- Last but not least, irrigated crops usually appear greener and lush compared with near crops.

We added this in the GVG data description in Line 186-189:

*Also, irrigated was applied for certain crop types, such as winter wheat in North China Plain, Cotton in Xinjiang and vegetable and tomatoes in most province, et.al. Meanwhile, irrigated crops usually appear greener and lush compared with near crops. Even it cannot be distinguished following above characteristics, the injury of local farmer could give the answer.*

We also include above information in the supplementary materials.

**Q4.** The irrigation map and GCPIS were identified using two ways (irrigation performance and DL), but some figures make me confused to display these two results.

Response: Thanks for your valuable comments.

The irrigation map was identified using irrigation performance while the irrigation method, specifically for central pivot irrigation system, was identified using DL method. As for the figures we have changed the display manner in the MS for Figure1, 6, 16.

Please see the following response in detail.

**Minor revision:**

**Q1.** The preprocess of NDVI data in Line 160 should be further explained.

Response: Thanks for your suggestion.

We added more explanations in the text to describe the preprocessing to NDVI data.

Line 165-166:

*The 30-metre spatial resolution NDVI data from the Landsat sensors Thematic Mapper (TM), Enhanced Thematic Mapper Plus (ETM+), and Thermal Infrared Sensor (OLI-TIRS) onboard Landsat-5, Landsat-7, and Landsat-8, respectively, were utilized in Google Earth Engine (GEE) (Gorelick et al., 2017) to differentiate irrigated and nonirrigated areas across various IMZs during a specific period. The NDVI data was masked using the cloud and water mask in the flag file and rescaled into the same range between -1 and 1.*

**Q2.** You could list some detailed maps of global CPIS in Figure 6 to make the global CPIS map clearer.

Response: Thanks for your suggestion.

We added the detail map of CPIS in Figure6. Figure b-d are the detail map of CPIS. The location of each sub figure was labelled in the main global map.

[Figure]

**Q3.** In Figure 16 it will be significant if the satellite images were added to give the reader a basis for their judgment.

Response: Thanks for your suggestion.

We have revised the figure accordingly.

[Figure]

**Q4.** IMZ was not so readable in Figure 1.

Response: Thanks for your comment.

We have separate figure one as two to make the element such as IMZ boundary clearer.

[Figure]

[Figure]

b) S12W48    c) S36W102    d) S48W114    e) N24E42    f) N24E24

**Q5.** The English should be further polished and improved.

Response: Thanks for your suggestion.

we have polished our MS and the certification is shown as below.

[Figure]

**RC3**

This manuscript introduced the GMIE-100 dataset, which identifies global irrigated cropland using drought stress performance and machine learning. This is a valuable dataset that could benefit various fields, including agriculture, environmental science, and water resource management. However, I have some major concerns about this MS that need the authors to clarify before it is further processed.

**Q1.** The title of the manuscript indicates that the dataset represents the largest irrigated area. How does the author interpret this "largest area"? This requires the author to provide explicit clarification within the text. Additionally, how does the author consider the possibility of overestimation of this largest area relative to the actual distribution, given that our focus is on the actual distribution range?

Response: Thanks for your valuable suggestion comments.

The largest area should be understood separately for region with regular irrigation (RIR)and region with irregular irrigation (RIO). For RIR, the largest area means the cropland area irrigated one time at least for last three years (2017-2019). Because we detect irrigation every year for this region. To avoid missing fallow land, we identify the largest extent for last three years (2017-2019).

For RIO, it means the cropland area irrigated one time at least for last ten years (2010-2019). For RIO, irrigation occurs occasionally. We detect whether the cropland is irrigated in the driest year. But in the normal year, the irrigation maybe not necessary in this area. So, this means the largest extent area for last ten years (2010-2019).

We add this explanation in the conclusion and discussion part.

Line 504-511:

*As for the maximum extent should be understood separately for RIR and RIO. For RIR, the largest area means the cropland area irrigated one time at least for last three years (2017-2019). Because we detect irrigation every year for this region. To avoid missing fallow land, we identify the largest extent for last three years (2017-2019). For RIO, it means the cropland area irrigated one time at least for last ten years (2010-2019). For RIO, irrigation occurs occasionally. We detect weather the cropland is irrigated in the driest year. But in the normal year, the irrigation maybe not necessary in this area. So, this means the largest extent area for last ten years (2010-2019). On the other hand, when we compare our result with nation census data, the result shows high consistent. Compared with USGS-LGRIP30 and GRIPC-500, our result didn't show much overestimation.*

**Q2.** The samples are derived from different collection methods. It is crucial for the author to clarify whether samples collected through different methods exhibit consistent

representation and describe irrigated land in the same manner. If their collection standards vary, the author needs to explicitly discuss the impact on the results.

Response: Thanks for your valuable suggestion.

The representation of samples was extremely important for the final accuracy. Nevertheless, it is hard to collect the irrigation field point globally, even crop types samples. So, we fused three independent sources, the GVG field data, USGS-samples and visual interpenetration data. You can see the distribution of samples from three sources in the following figures and a specific number for each country.

[Figure]

Table S3 Number of samples in different countries and sources

| Sources | Number | Distributed country |
|---|---|---|
| GVG field data | 78,338 | China(72,224) \Cambodia\Ethiopia\Zambia\ Zimbabwe |
| USGS-samples | 17,076 | Brazil (13,368), Australia (2,192), Thailand (393), and Tunisia (389) |
| VHR-interpratation | 19,965 | Rest Countries |
| total | 115,379 | |

From different country, there is varied dominant samples source. Such as in China, most of samples was obtained from GVG field survey. While in Brazil, major samples were from USGS samples. Except country with GVG and USGS-samples, the visual interpretation data was dominant sources of samples. This also ensure the represented manner of irrigated cropland.

This could definitely introduce some uncertainty in terms of samples representatives. This effect should be acceptable in arid and semi-arid regions because the irrigation performance

is relatively easy to identify. However, the uncertainty maybe enlarged in wet region due to complex manner of irrigated cropland.

We add this uncertainty of representations in the discussion part

Line 558-567:

*"Thirdly, it is hard to collect the filed samples globally, we fused three sources of samples. From different country, there is varied dominant samples source. Such as in China, most of samples was obtained from GVG field survey. While in Brazil, major samples were from USGS samples. Except country with GVG and USGS-samples, the visual interpretation data was dominant sources of samples. This also ensure the represented manner of irrigated cropland. Overall, the number of samples was very large. Basically, this irrigated and rain-fed samples database could meet the globally irrigated cropland mapping compared with global cropland expansion mapping research (Potapov et al., 2022), which achieved cropland mapping globally with thousands of samples. Meanwhile, this fused samples maybe introduce some uncertainty in terms of representation. This effect should be acceptable in arid and semi-arid regions because the irrigation performance is relatively easy to identify. However, the uncertainty maybe enlarged in wet region due to complex manner of irrigated cropland. "*

**Q3.** In terms of accuracy assessment, merely providing overall accuracy is insufficient. Please refer to best practices for reporting accuracy as outlined in papers such as Olofsson et al. 2014 [1]. Moreover, I have not observed quantification of uncertainty, which necessitates further work from the author.

Olofsson P, Foody GM, Herold M, Stehman SV, Woodcock CE, Wulder MA. Good practices for estimating area and assessing accuracy of land change. Remote Sensing of Environment 2014; 148:42–57. https://doi.org/10.1016/j.rse.2014.02.015.

Response: Thanks for your valuable comments.

We changed all the accuracy assessment following the commended practice and evaluate the uncertainty of total area estimation.

Briefly, the overall accuracy of GIME-100 was 83.6%±0.6% with producer accuracy of 86.1%±0.7% and UA of 82.20%±0.8%. And the total area of GMIE is estimated as 403.17±9.82Mha, accounting for 23.4%±0.6% of the global cropland.

For the GCPIS data, the overall Accuracy was 97.87%±0.1% with producer accuracy of 81.75%±0.2% and UA of 92.68%±0.1%. And the total area of GCPIS is estimated as 11.5±0.01Mha.

We have changed the statement of accuracy assessment and area estimation in the body text.

**Q4.** The results and discussion sections lack necessary citations. Many explanations proposed by the author lack corresponding literature support, which makes it difficult for me to be convinced of the correctness of your interpretations. Please see the annotations I've made in the manuscript.

Response: Thanks for your specific comments. We add necessary citation in the revised version.

**Q5.** I have made several annotations in the manuscript indicating areas that need revision. It is advised that the author make corresponding modifications and carefully review the entire document to rectify similar errors.

Response: Thanks for your nice suggestion.

Firstly, AJE have re-polished this MS for us, and the certification is show as below. Also, we carefully check the whole MS again and revised the similar errors.

---

## Referee Report (RR1)

[referee-annotated manuscript omitted]

---

## Author Response (AR2)

We are highly appreciated for your constructive comments and suggestions on our manuscript. Those comments and suggestions are valuable and helpful for revising and improving our article, as well as inspiring our research. We have carefully reviewed the comments and have revised the manuscript accordingly. Our responses are given in a point-by-point manner below and **BLUE** fonts. Please find our detailed responses in supplement to all these comments/suggestions and thank you again for everything you have contributed.

**RC1**

Overall, this will be a useful work to publish. However, there are, currently, numerous highly significant issues that needs to be addressed and re-reviewed before further consideration.

1. Very poor reference to literature

I have marked at number of places where existing Landsat-derived rainfed and irrigated area product @ 30 m (LGRIP30) is completely overlooked. Authors claim other existing products are coarse but fail to mention at number of places LGRIP30 which is a 30 m global irrigated and rainfed cropland product exists.

Apart from that reference to literature pertaining to previous irrigated and rainfed areas is poor. I definitely like to see upfront reporting of this keeping in view highest scientific ethics.

**Response:**

Thanks for your useful comments we included the Landsat-derived rainfed and irrigated area product at 30 meters (LGRIP30) in the list of existing products. We acknowledge the oversight in our initial submission and have now added a detailed description of LGRIP30 within the text.

We would like to clarify that our product GMIE, was published in March 2023, coinciding with the release of LGRIP-30. This timing may have contributed to the initial lack of a comprehensive description of LGRIP-30 in our introduction. However, we have since rectified this by including a thorough introduction to LGRIP-30, recognizing its importance as a high-resolution irrigated land product. We have also taken the opportunity to compare our GMIE with LGRIP-30, highlighting the similarities and differences in methodology, application, and results. This comparison is aimed at providing a clearer understanding of the contributions our work makes to the field of high-resolution agricultural land classification using remote sensing.

We appreciate the reviewer's guidance in maintaining the highest scientific standards and ethics in our work. We believe that the revisions made will strengthen the manuscript and ensure that it acknowledges and builds upon the existing body of research in a respectful and scholarly manner.

Line 92-98:

*Among these data, the Landsat-derived Global Rainfed and Irrigated-area Product (LGRIP30) is a high-resolution irrigated cropland with an overall accuracy of 86.5% using advanced machine learning algorithms, which is released on Feb 2023 and available through NASA's Land Processes Distributed Active Archive Center (LP DAAC)(Teluguntla et al., 2023). The LGRIP30 data indicates a total global net irrigated area (TGNIA) of 0.71 billion hectares among all cropland area of 1.80 billion hectares of croplands, ie the irrigation proportion was about 39.44%, suggesting a notably*

*high proportion compared with exiting result (Thenkabail et al., 2009; Siebert et al., 2015).*

2. Methods

Authors divide the world into 110 zones and use very simplistic NDVI approach to determining where are the irrigated and rainfed croplands. This approach when applied within each zone like those in Indus or Ganges will provide reasonable results in separating irrigation from rainfed. But in numerous other zones where there is minor ground water irrigation or in humid areas will have huge uncertainties.

Why is such a simplified approach adopted for such a complex problem of separating irrigated from rainfed. Please refer to algorithm theoretical basis document (ATBD) of LGRIP30:

https://lpdaac.usgs.gov/products/lgrip30v001/

I understand that such a simplified approach is easy to code in GEE. But, uncertainties of the outputs will be huge.

**Response:**

Thank you for your valuable comment. We acknowledge that decision tree methods are indeed beneficial for information mining and can handle complex datasets to extract useful insights. Nevertheless, In developing our method, we have retraced the essence of irrigation to identify key time windows that require irrigation. We use these time windows along with vegetation indices to differentiate between irrigated and rainfed croplands. While the final thresholding approach is relatively straightforward to implement in Google Earth Engine (GEE), the selection of appropriate time windows, preparation of multi-year remote sensing data, and accurate zoning processes are crucial.

As you pointed out, the accuracy of our method does indeed vary across different regions. This is due to the variability in climate, soil types, crop species, and irrigation practices among different areas. According to the accuracy reports for each irrigation mapping zone, there is cropland in 105 zones of total 110 irrigation mapping zones,whilw 96 of them have an accuracy greater than 70%. There are just 9 divisions with accuracy less than 70%, most of which are located in the Southeast Asian Island countries, regions like Thailand, Myanmar, Laos, and the tropical rainforest areas of South America (Amazon), which are humid regions.

Furthermore, our method has been published in the journal *Global Environmental Change (GEC)*, where the article provides a detailed description of our approach and implementation process. The current paper is a description of the dataset generated by that method. Our goal is to provide a reliable dataset for researchers and policymakers to better understand and manage irrigated croplands.

We recognized that there is huge uncertainty in the above-mentioned regions, but the irrigation proportion in this region is usually not that much compared with arid and semi-arid regions. Meanwhile, the identification of irrigation in these regions using the machine leaning methods is also challenging task and not easy to fully distinguish irrigated and rainfed cropland without proper feature inputs.

We understand and agree with your concerns about the uncertainties that may arise for humid regions. Therefore, we have also discussed these potential uncertainties in the paper and suggested possible directions for improvement in future research. We believe that the combination of irrigation performance assessment to choose optimal time windows and powerful machine learning methods could be potential way to handle this problem for humid regions.

Line 563-573:

*The accuracy of our method indeed varies across different regions due to the variability in climate, soil types, crop species, and irrigation practices among different areas. According to the accuracy reports for each irrigation mapping zone, cropland is present in 105 out of the total 110 irrigation mapping zones, with 96 of them exhibiting an accuracy greater than 70%. There are only 9 divisions with accuracy less than 70%, most of which are situated in the Southeast Asian island countries, regions such as Thailand, Myanmar, Laos, and the tropical rainforest areas of South America, notably the Amazon, which are characterized by their humid conditions. We acknowledge that there is significant uncertainty in these aforementioned regions; however, the proportion of irrigation in these areas is typically not as substantial compared to arid and semi-arid regions. The task of identifying irrigation in these regions using machine learning methods is also challenging, as it is not straightforward to fully distinguish between irrigated and rainfed cropland without accurate phenological inputs. A potential solution for improving accuracy in humid regions could involve the integration of irrigation performance assessments to select optimal time windows, coupled with advanced machine learning techniques.*

4. Accuracy assessments
Each of the 110 zones much have accuracy error matrices.
**Response:**
  Thank you for your comment regarding the accuracy assessments for each of the 110 zones in our study. We understand the importance of providing detailed accuracy error matrices for each zone to ensure the credibility and robustness of our findings.
  In response to your request, we have prepared and included detailed accuracy reports for each zone as supplementary material. These reports contain point number used for validation and overall accuracy for each zone, which are useful for understanding the reliability of the data.
  We appreciate your comments and look forward to your further feedback on our revised manuscript and provided supplementary materials.

Line 434:

*The specific accuracy for each IMZ could refer to Table S1*

(Table — regional cropland and climate data; columns: Region code, Region name, Cropland Sources &year, Crop Types, Climate Zone, Driest Year, Dry Months, P, P/PET, P-PET, Threshold (NDVI at Dry season), NDVI deviation at Extreme events, NDVI difference (average irrigated – average rainfed), Overall accuracy, Point number used for validation)

5. cropland mask

Also refer to this important work on global cropland mask:

https://lpdaac.usgs.gov/news/release-of-gfsad-30-meter-cropland-extent-products/

**Response:**

Thank you for drawing our attention to the important work on global cropland mask released by the Land Processes Distributed Active Archive Center (LP DAAC). We appreciate your suggestion to incorporate this valuable resource into our study.

The Global Food Security-support Analysis Data (GFSAD30) offers invaluable high-resolution data on cropland extent worldwide, which is essential for informed decision-making in areas such as water sustainability and food security. Actually, we have already integrated GFSAD30 data into our synthesized cropland mask for Southeast Asia. However, due to varying definitions of what constitutes cropland, we have not applied this data in other regions. Our focus has been primarily on seasonal croplands, as permanent crops—such as fruit and nut trees, as well as coffee, tea, and certain vines—are often classified as shrubland or tree cover in most land cover classification systems. Nevertheless, it's important to note that the GFSAD30 includes these continuous plantations within its cropland data. (Phalke, Özdoğan et al. 2020).

We will ensure that the GFSAD30 data is properly cited in our manuscript. Thank you once again for your valuable feedback, which undoubtedly enhances the quality and integrity of our work.

*Line 241:*

*Thenkabail, P.S., Teluguntla, P.G., Xiong, J., Oliphant, A., Congalton, R.G., Ozdogan, M., Gumma,*

*M.K., Tilton, J.C., Giri, C., Milesi, C., Phalke, A., Massey, R., Yadav, K., Sankey, T., Zhong, Y., Aneece, I., and Foley, D., 2021, Global Cropland-Extent Product at 30-m Resolution (GCEP30) Derived from Landsat Satellite Time-Series Data for the Year 2015 Using Multiple Machine-Learning Algorithms on Google Earth Engine Cloud: U.S. Geological Survey Professional Paper 1868, 63 p., https://doi.org/10.3133/pp1868.*

6. Definitions

What is irrigated areas?. Do you consider an area as irrigated if it gets water once in growing season or is the area irrigated if it is irrigated during one season and not the other. Definitions are key to mapping. But, clarity is lacking.

**Response:**

Thank you for your inquiry about the definition of irrigated areas within our study. Your point about the importance of clear definitions for accurate mapping is well-taken.

Irrigated cropland is characterized as agricultural land that benefits from human interventions and equipped with irrigation infrastructure, including facilities like canals and central pivot systems. In our study, an irrigated area is defined as a land area where water is artificially supplied to the crops at least once during the growing season to supplement natural rainfall. This definition includes areas that receive irrigation at any time during the season, regardless of whether they are irrigated in every season or not.

Therefore, we have revised our manuscript to include a more explicit definition of irrigated areas. This definition will be clearly stated in the methods section to ensure that there is no ambiguity for readers and users of our data. We appreciate your feedback and the opportunity to clarify our methodology. We believe that these revisions will enhance the quality and precision of our research.

Line 133-136:

*So, the Irrigated cropland is characterized as agricultural land that benefits from human interventions and is outfitted with irrigation infrastructure, including facilities like canals and central pivot systems(Salmon et al., 2015; Meier et al., 2018). This definition includes areas that receive irrigation at any time during the season, regardless of whether they are irrigated in every season or not.*

7. area calculations

Only net irrigated areas are calculated. What about gross irrigated areas? In same piece of land crops are grown one, two, or three times in some areas. How do you distinguish that.

**Response:**

Thank you for your insightful question regarding the calculation of irrigated areas, specifically the distinction between net and gross irrigated areas and the management of multiple cropping cycles within the same piece of land.

We have concentrated on the net irrigated area, which represents the actual land area equipped and utilized for crop irrigation. This approach is commonly used to assess the land area that requires water resources for irrigation purposes. However, gross irrigated cropland area encompasses all the land that could be irrigated during a crop's growing season, regardless of whether it is continuously irrigated throughout the season. For instance, if a plot of land is planted and irrigated twice in one

growing season, that land would be counted twice, reflecting in the gross irrigated cropland area. Therefore, the gross irrigated area may exceed the net irrigated area because it accounts for instances of multiple plantings and irrigations. This distinction is vital for accurately assessing the use of water resources and planning agricultural production.

In our research, we estimate maximum irrigation extent under the assumption that irrigation equipment is primarily deployed to mitigate the most water-stressed conditions (such as the dry season in the RIR and extreme drought events within ten years for the RIO). Regarding multiple cropping cycles, our methodology identifies an area as irrigated if irrigation occurs at least once within a crop season. For the regions need regular irrigation (RIR), we choose only the dry season & growing season that experiences the greatest water stress for every year to estimate the net irrigation in that growing. Similarly, for the region needs irrigation only occasionally for some years (RIO), we evaluate net irrigation area based on a single growing season that has undergone an extreme drought event in the last decade. After all, we didn't consider the multiple cropping with in one-piece land. So, we just estimate the net irrigation area for selected growing season, whose value should be largest during that decades or three years.

We have expanded our discussion in the manuscript to include a more comprehensive analysis of both net and gross irrigated cropland, as well as future perspectives on this topic.

Line 525-536:

*When discussing irrigation extents, it is crucial to differentiate between "net irrigated area" and "gross irrigated cropland area." The net irrigated area refers to the actual land area equipped with irrigation facilities and receiving irrigation, while the gross irrigated cropland area encompasses all the land that could be irrigated during a crop's growing season, regardless of whether it is continuously irrigated throughout the season. For instance, if a plot of land is planted and irrigated twice in one growing season, that land would be counted twice, reflecting in the gross irrigated cropland area. Therefore, the gross irrigated area may exceed the net irrigated area because it accounts for instances of multiple plantings and irrigations. This distinction is vital for accurately assessing the use of water resources and planning agricultural production. In our research, we estimate maximum irrigation extent under the assumption that irrigation equipment is primarily deployed to mitigate the most water-stressed conditions. So, we just estimate the net irrigation area for selected growing season, whose value should be largest during that decades or three years. For RIR, we estimate the net irrigation in the dry season & growing season that experiences the greatest water stress for every year. Similarly, for RIO, we evaluate net irrigation area based on a single growing season that has undergone an extreme drought event in the last decade.*

8. Uncertainties in irrigated area map and area calculations
So, if the proportion of a pixel irrigated is say 10%., so you then only calculate fraction of the pixel area as irrigated or is it full pixel area. This is unclear.
**Response:**
Thank you for addressing the uncertainties in our irrigated area map and the calculations therein. We appreciate your emphasis on the necessity for precise definitions to ensure the accuracy of our mapping efforts.

In our methodology, a parcel of land is designated as irrigated if it receives any supplemental artificial water supply to support crop cultivation at least once during the growing season. The Global Maximum Irrigated Extent (GMIE) dataset, initially developed at a 30-meter resolution, categorizes each pixel as either irrigated or rainfed cropland. Thus, if a pixel contains at least 10% irrigated cropland, it is classified as an irrigated pixel within that 30×30 meter area. We recognize that the actual extent of irrigation at 30m resolution can fluctuate due to factors such as crop rotation and the presence of fallow land, which are clearly discernible at the 30-meter resolution and can influence the overall measurement of irrigated cropland. To mitigate these variations and enhance the accuracy of our data, we have calculated the proportion of irrigated cropland within a larger 100 m × 100 m grid.

As the result, there may be a tendency towards overestimation due to the mixed pixels at the 30-meter resolution, particularly in regions with smaller fields such as Southern China, Southeast Asia, and parts of Africa. However, the relatively high resolution of the pixels helps to mitigate this uncertainty to a certain extent.

We added more discussion regarding to this uncertainty of overestimation.

Line 600-607:

*Also, a parcel of land is designated as irrigated if it receives any supplemental artificial water supply to support crop cultivation at least once during the growing season. The Global Maximum Irrigated Extent (GMIE) dataset, initially developed at a 30-meter resolution, categorizes each pixel as either irrigated or rainfed cropland. Thus, even if a pixel contains less than 100% irrigated cropland, it is classified as an irrigated pixel within that 30×30-meter area. As the result, there may be a tendency towards overestimation due to the mixed pixels at the 30-meter resolution, particularly in regions with smaller fields such as Southern China, Southeast Asia, and parts of Africa. However, the relatively high resolution of the pixels helps to mitigate this uncertainty to a certain extent.*

9. Irrigation Method
There are numerous types of irrigation. Centre Pivot irrigation is well mapped. However, rest are all totally unclear. I suggest this aspect is completely removed from the manuscript and the manuscript is limited to irrigated and rainfed.
**Response:**
Thank you for your insightful comments and suggestions regarding the manuscript. We have given careful consideration to your recommendation to remove the sections discussing various types of irrigation systems that are not well-documented or clear, and to focus the manuscript on irrigated and rainfed systems.

After thorough evaluation, we have decided to retain the section on Centre Pivot irrigation. Our rationale for this decision is based on the fact that Centre Pivot irrigation is one of the most efficient and widely used systems globally. Moreover, we have identified a significant gap in the global mapping of CPIS, although there is some research mapping the CPIS for the dryland (Chen, Zhao et al. 2023). In light of this, we propose to maintain the current scope of the manuscript, which includes the part and description of global Centre Pivot irrigation, and to emphasize the importance of further research and data collection on other types of irrigation systems.

In another hand, we changed the title to "GMIE-100: A global maximum irrigation extent and central pivot irrigation system dataset derived via irrigation performance during drought stress and machine learning methods" to deal with the problem that other irrigation type is not well documented. Also, in the discussion part, we put these points in Limitation and outlook the identification of other irrigation types in the future with the help of big-geo data, which is important for water use estimations.

Line547-550:

*However, this study didn't include the lateral other irrigation types, because the identification of irrigation CPIS method was relied on the circle shape in the satellite data and other irrigation typesthe lateral irrigation didn't show this distinguish feature. The identification of other irrigation types in the future is definitely important for water use estimations (Boutsioukis and Arias-Moliz, 2022), maybe with the help of big-geo data.*

1) Chen, F., H. Zhao, D. Roberts, T. Van de Voorde, O. Batelaan, T. Fan and W. Xu (2023). "Mapping center pivot irrigation systems in global arid regions using instance segmentation and analyzing their spatial relationship with freshwater resources." Remote Sensing of Environment 297: 113760.
2) Phalke, A. R., M. Özdoğan, P. S. Thenkabail, T. Erickson, N. Gorelick, K. Yadav and R. G. Congalton (2020). "Mapping croplands of Europe, Middle East, Russia, and Central Asia using Landsat, Random Forest, and Google Earth Engine." ISPRS Journal of Photogrammetry and Remote Sensing 167: 104-122.

---

## Author Response (AR3)

**Public justification (visible to the public if the article is accepted and published)**:
Once again, this study filled an important data gap in mapping crop management, and the reported dataset nicely complements other newly and under-development regional or global products such as LGRIP30. The manuscript has been reviewed by four experts in this area, unfortunately not all of them are able to follow through the whole process and re-review how the authors' revision addressed their original concerns. But I thoroughly went through the responses and found the authors adequately addressed all the comments raised (especially by the fourth reviewer). With that said, I am happy to recommend Accept, contingent on a minor edit I suggested below.

Response: Thank you for your thorough evaluation of our manuscript and for considering the feedback from all reviewers. We are pleased to hear that you find our study to be a valuable contribution to the field of crop management mapping and that it complements other regional or global products such as LGRIP30.
We appreciate the time and effort you have invested in ensuring that all reviewer comments were adequately addressed in our revisions. We understand the importance of a rigorous review process and are grateful for the expertise provided by the reviewers, even if not all could follow through to the end.
Regarding the minor edit you have suggested, we are more than willing to make the necessary changes.

**Additional private note (visible to authors and reviewers only):**
I don't really have any technical concerns, but for any scientific writing, there is always room to improve for better clarity. At least, I suggest the authors to consider revising the title a little bit. For example, what does the 100 in "GMIE-100" mean? If that is an important piece of info, it is better to explicitly spell out the resolution in the title. Machine learning methods are not too different from machine learning; even better, it may be more useful to just directly say "deep learning" rather than machine learning methods.

Response: Thank you for your suggestion on improving the clarity of our scientific writing. We have revised the title to "GMIE: A Global Maximum Irrigation Extent and Central Pivot Irrigation System Dataset Derived via Irrigation Performance During Drought Stress and Deep Learning Methods." to eliminates any ambiguity regarding the "100" and emphasizes the use of deep learning methods.

Last but not the least, I highly encouraged the readers to proofread the manuscript a few times to correct for any potential awkward use of English. Here is an example showing what I meat: In the abstract, it says " To our knowledge, this study is the first attempt to identify irrigation methods globally". It makes little sense to say "identify identify irrigation methods globally". Do you mean "mapping irrigation globally" or "developing global-scale irrigation-mapping algorithms"?

Response: We appreciate your attention to detail and agree that clear and precise language is crucial for effective communication of our research.

Upon reviewing the sentence you mentioned, we recognize that the phrase "identify irrigation methods globally" may not accurately reflect our intended meaning. We have revised the sentence in the abstract to state: "To our knowledge, this is the inaugural study to undertake a global identification of specific irrigation methods, with a focus on the CPIS." This distinction is crucial as understanding the global distribution of irrigation types allows for a more precise estimation of irrigation efficiency. Due to the variation in irrigation efficiency among for different types of irrigation methods, CPIS demonstrate an efficiency exceeding 80%, while gravity-flowing irrigation methods exhibit a comparatively low efficiency of approximately 60% (Waller and Yitayew, 2016). Therefore, irrigation efficiency can be estimated based onin relation to types of irrigation methods in the future. Recognizing this, our study aims to enhance the accuracy of water resource management by mapping these irrigation methods on a global scale.

Additionally, a native English speaker has helped us polish the manuscript once again, making necessary revisions. We have proofread the entire manuscript to avoid minor mistakes or misunderstandings. Please review the changes in the manuscript with tracked changes enabled.

---

## Author Response (AR4)

With the next file upload request, please consider adding supplement`s captions to photos in the supplement. Please see more: https://www.earth-system-science-data.net/submission.html#assets > Supplements.

Response: Thank you for your guidance on the submission process and the specific instructions regarding the supplements for our manuscript. We have carefully reviewed the section on "Supplements" at the provided link: ESSD - Submission, and we understand the importance of including captions for photos within the supplement.

We ensure that the supplement file is accompanied by clear and descriptive captions for each image. Please see the revised supplement.